# Resolving the structural basis of therapeutic antibody function in cancer immunotherapy with RESI

Isabelle Pachmayr [1,2], Luciano A. Masullo [1], Susanne C. M. Reinhardt [1,3], Jisoo Kwon[1], Maite Llop [4], Ondřej Skořepa [1,5], Sylvia Herter [4], Marina Bacac[4], Christian Klein [2,4] & Ralf Jungmann [1,3] ✉

Monoclonal antibodies (mAb) are key therapeutic agents in cancer immunotherapy and exert their effects through Fc receptor-dependent and -independent mechanisms. However, the nanoscale receptor reorganization resulting from mAb binding and its implications for the therapeutic mode of action remain poorly understood. Here, we present a multi-target 3D RESI super-resolution microscopy technique that directly visualizes the structural organization of CD20 receptors and the Type I (e.g., Rituximab) and Type II (e.g., Obinutuzumab) anti-CD20 therapeutic antibodies and quantitatively analyze these interactions at single-protein resolution in situ. We discover that, while Type I mAbs promote higher-order CD20 oligomerization, Type II mAbs induce limited clustering, leading to differences in therapeutic function. Correlating RESI with functional studies for Type II antibodies with different hinge region flexibilities, we show that the oligomeric CD20 arrangement determines the Type I or Type II function. Thus, the nanoscale characterization of CD20-mAb complexes enhances our understanding of the structure-function relationships of therapeutic antibodies and offers insights into the design of next-generation mAb therapies.

Immunotherapies based on therapeutic monoclonal antibodies (mAb) have revolutionized cancer treatment in the last 30 years. Tumor cell killing by mAbs is mediated by various mechanisms: On the one hand, Fc recognition by cellular Fc receptors can lead to antibody-dependent cellular cytotoxicity and phagocytosis, and Fc recognition by soluble complement proteins can activate complement-dependent cytotoxicity (Fig. 1a). On the other hand, binding of mAbs to their membrane receptor targets can directly exert function on the treated cells, independent of Fc functionality (Fig. 1a)[1,2]. In both the Fc-dependent and the Fc-independent mechanisms, the nanoscale receptor reorganization caused by mAb binding has likely a strong impact on their downstream function. This is thought to be the case for tumor necrosis factor receptor

mAbs, for which mAbs exert agonism or antagonism, depending on their clustering behavior, as well as for growth factor receptors, such as EGFR and C-Met, in which different mAb clones either lead to activation or silencing[3–6].

A prominent example of this hypothesized mAb structure-function relationship are anti-CD20-mAbs, used for treatment in B-cell malignancies and depletion of B cells in autoimmune diseases[7]. CD20-targeting mAbs can be separated into two groups—Type I and Type II—that exert therapeutic functions to a different degree: Type I mAbs lead to CD20 clustering and complement activation, whereas Type II mAbs lead to potent effector cell-mediated killing as well as to direct cytotoxicity[8,9]. Cryo-EM structures indicate a relationship of structure and function for Type I vs. Type II mAbs: Type I Rituximab

[1]Max Planck Institute of Biochemistry, Planegg, Germany. [2]Department of Biochemistry, Ludwig Maximilian University, Munich, Germany. [3]Faculty of Physics and Center for Nanoscience, Ludwig Maximilian University, Munich, Germany. [4]Roche Innovation Center Zurich, Roche Pharma and Early Development, Schlieren, Switzerland. [5]Department of Biochemistry, Charles University, Prague, Czech Republic. ✉e-mail: jungmann@biochem.mpg.de

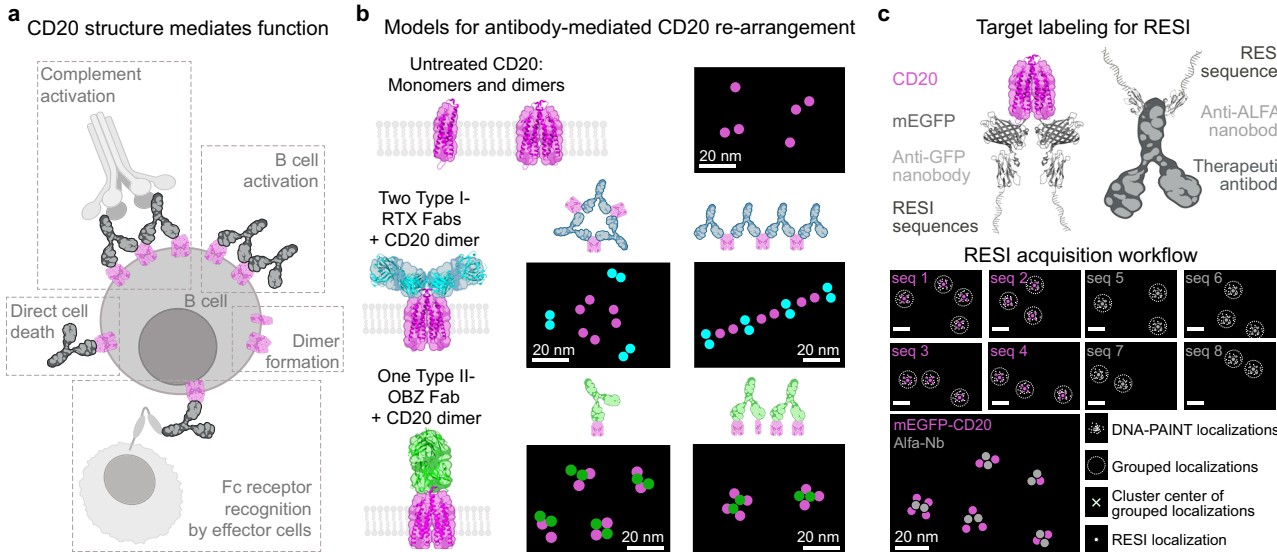

**Fig. 1 | Assessing the structure-function relationship of anti-CD20 therapeutic antibodies in the cellular context using RESI microscopy. a** The structural configuration of CD20 proteins and their interactions with therapeutic antibodies on the cell membrane influence the therapeutic efficacy of the antibodies. **b** CD20 proteins (magenta) in the membrane exist as a mixture of monomers and pre-formed dimers[1]. Rituximab (RTX) (cyan) can bind with 2 Fabs per CD20 dimer, thereby inducing higher-order arrangements of CD20 dimers[2,3]. However, the quantitative nature of the 3D cluster organization is still unknown. In contrast, Obinutuzumab (OBZ) (green) can bind with one Fab per single CD20 dimer, suggesting a terminal complex of up to CD20 tetramers by bridging two CD20 dimers.

However, the resulting structural organization of CD20 – potentially forming monomers, dimers, trimers, or tetramers – remains to be fully elucidated. **c** We use two-target RESI super-resolution microscopy with four rounds per target to visualize and correlate the locations of mEGFP-tagged CD20 and ALFA-tagged therapeutic antibodies at single-protein resolution (sub-5 nm). CD20 is labeled in 1:1 stoichiometry and mAbs are labeled in a 2:1 stoichiometry. By performing eight consecutive DNA-PAINT imaging rounds and clustering localizations, we achieve precise RESI localizations ($\sigma \approx 0.6$ nm) of the targets in their cellular context. Created with the help of BioRender (https://BioRender.com/y76v9f4 and https://BioRender.com/5575210).

(RTX) fragment antigen-binding (Fab) exhibit a shallow binding angle and a 2:2 Fab:CD20 stoichiometry, suggesting a potential bridging of CD20 dimers upon Type I mAb treatment (Fig. 1b, top). In contrast, Type II Obinutuzumab (OBZ) Fabs have a steeper CD20 binding angle and a 1:2 Fab:CD20 stoichiometry[10], suggesting a maximum of tetramers after Type II mAb binding (Fig. 1b, bottom)[11,12].

However, for both Type I and Type II treatment, the nanoscale structural arrangement of CD20 together with the complete mAbs in a cellular environment is not fully understood. Additionally, there is a lack of insight into the nanoscale organizational requirements for Type I or Type II function. To address these questions, we need (1) molecular specificity, (2) the ability to image proteins in the context of intact cells, (3) molecular spatial resolution and (4) multiplexing capability. Cryo-EM, mass spectrometry and traditional super-resolution microscopy methods, such as stimulated emission depletion microscopy, stochastic optical reconstruction microscopy (STORM) or photo-activated localization microscopy, are limited because they cannot achieve single-protein resolution in situ for dense assemblies[13–15]. With the recent development of resolution enhancement by sequential imaging (RESI), we can achieve single-protein resolution in intact cells, complementing structural biology data in a cellular context[16]. Given the unresolved link between the nanoscale organization of CD20 and therapeutic functions, we aim to elucidate how therapeutic mAbs modulate CD20 spatial arrangements and how these, in turn, influence functional outcomes.

To this end, we introduce multi-target 3D RESI imaging to directly and simultaneously visualize the nanoscale organization of CD20 receptors and their bound therapeutic mAbs at the intact cell membrane. We find that Type I and Type II mAbs induce distinct receptor arrangements in situ, and that the degree of CD20 oligomerization correlates with a functional transition between the two mAb types. Our findings establish a generalizable framework for linking receptor nanoscale organization to the therapeutic antibody function and offer new avenues for optimizing antibody design. This approach paves the

way for structure-guided development of next-generation immunotherapies across diverse receptor systems.

## Results
### Single-protein resolution imaging of antibody-receptor complexes

To directly visualize both CD20 and the mAbs in the cellular context, we implemented a specific and quantitative multi-target labeling system for RESI. RESI is based on DNA-PAINT (DNA Points Accumulation for Imaging in Nanoscale Topography)[17,18], a super-resolution microscopy technique that utilizes the transient binding of fluorescently labeled DNA probes to complementary target sequences to achieve ~10 nm spatial resolution through single-molecule localization. By stochastically labeling and sequentially imaging sparse target subsets at this resolution, RESI allows us to enhance the precision of the DNA-PAINT measurements by averaging localizations, therefore achieving Ångström spatial resolution[16].

For CD20, this is achieved by tagging the receptors with monomeric enhanced GFP (mEGFP) and labeling with the cognate nanobody (GFP-Nb) (Fig. 1c, top left). For the therapeutic mAbs, we made use of a small and specific peptide tag (ALFA-tag)[19] that can be stoichiometrically labeled with its cognate nanobody and genetically encoded to be expressed at the C-terminus of the heavy chain of therapeutic antibodies RTX and OBZ (Fig. 1c, top right). To achieve RESI for two protein targets, we implemented stochastic labeling and sequential readout featuring two sets of four orthogonal DNA sequences to label CD20 and mAbs, respectively.

Each separate imaging round resulted in DNA-PAINT localizations, originating from repetitive detection of single molecules by stochastic blinking (Fig. 1c, bottom). Multiple DNA-PAINT localizations (K) were grouped according to spatial proximity ("clustered") to obtain RESI localizations with improved precision according to $\sigma_{RESI} = \sigma_{DNA-PAINT}/\sqrt{K}$. To achieve this, two adjacent molecules must be labeled with orthogonal docking strands. An image resolving

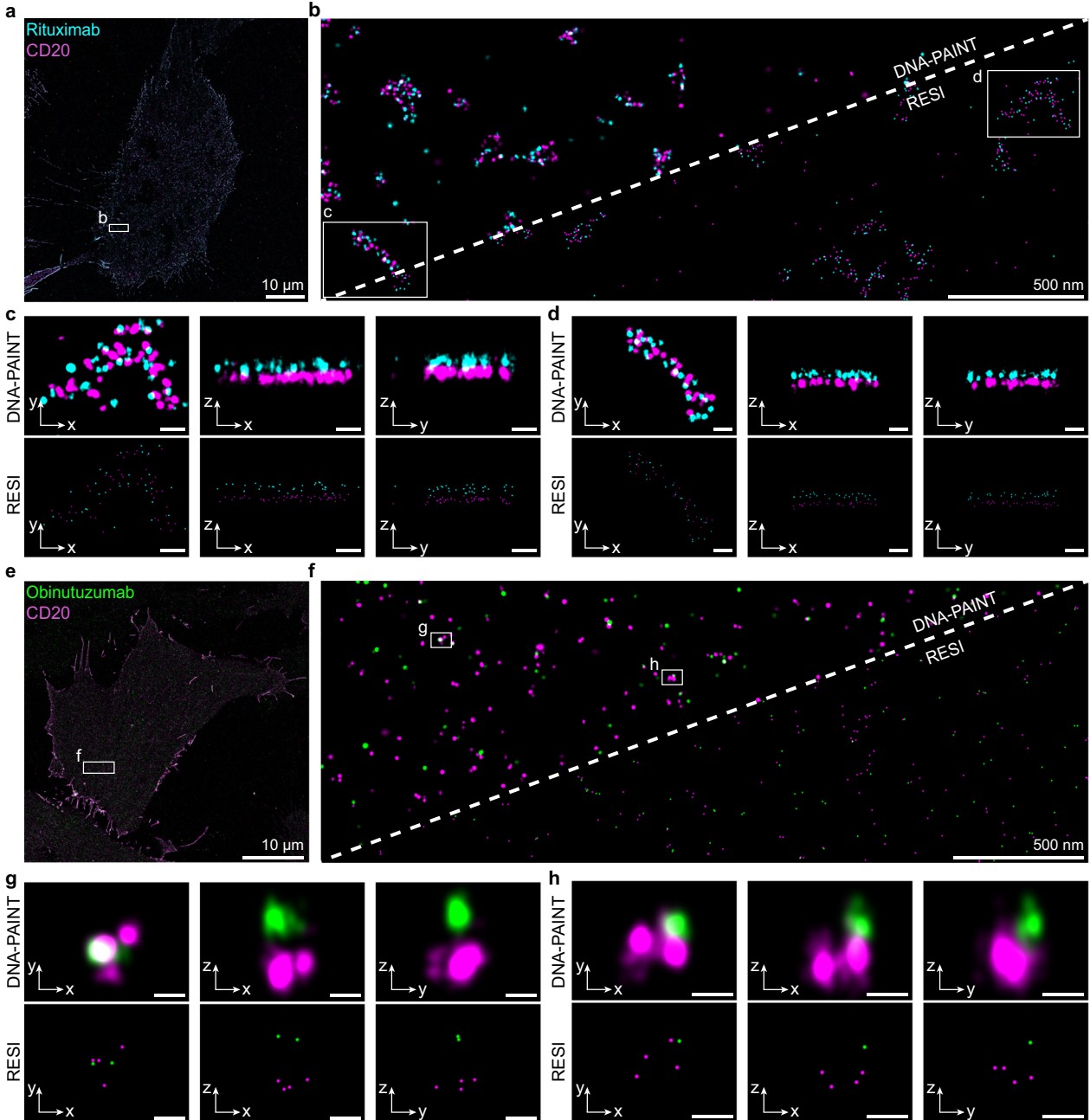

**Fig. 2 | Super-resolution imaging of CD20-therapeutic antibody complexes using RESI.** Imaging of cells treated with Rituximab (**a**) and Obinutuzumab (**e**) shows the distribution of CD20 and these therapeutic antibodies. Rituximab (cyan) colocalizes with CD20 (magenta), forming distinct higher-order structures on the cell membrane. In contrast, Obinutuzumab (green) and CD20 colocalize, yet appear homogeneously distributed without forming such structures. Transitioning from DNA-PAINT super-resolution (left side of dashed line) to RESI resolution (right side of dashed line) allows visualization of individual proteins in complexes. In (**b**), Rituximab-CD20 complexes exhibit clustered formations, while in (**f**),

Obinutuzumab-CD20 complexes appear as few colocalizing molecules without evident higher-order clustering. 3D rotational views of the complexes reveal further structural details. For Rituximab-CD20 complexes (**c**, **d**), the 3D view shows that clusters are generally planar, suggesting a two-dimensional organization in the cell membrane. In contrast, Obinutuzumab-CD20 complexes (**g**, **h**) show up to four CD20 molecules colocalized with a single Obinutuzumab molecule, lacking the distinct planar higher-order structures observed in Rituximab-treated cells. Images are representative of three independent experiments. Scale bars in (**c**, **d**): 50 nm. Scale bars in (**g**, **h**): 20 nm.

individual molecules was then reconstructed from the individual RESI localizations of CD20 and the mAb.

To observe the CD20 nanoscale organization independently of the downstream cellular functions in B lymphocytes, we established two-target RESI in CHO-K1 cells transfected with mEGFP-CD20 and treated with ALFA-tagged mAbs. After mAb treatment and fixation, we performed stochastic labeling according to Fig. 1c and performed whole-cell 3D RESI with 0.6 nm localization precision (Fig. 2 and

Supplementary Fig. 1). We achieved an effective resolution of ~2 to ~5 nm, limited by the size of the ALFA-tag-Nb or the mEGFP-tag-Nb complex, respectively.

## Type I and Type II antibodies show distinct nanoscale arrangements

Classical DNA-PAINT imaging shows that Type I-RTX and CD20 form co-clusters of 50-300 nm in the cell membrane (Fig. 2a, b). Only RESI,

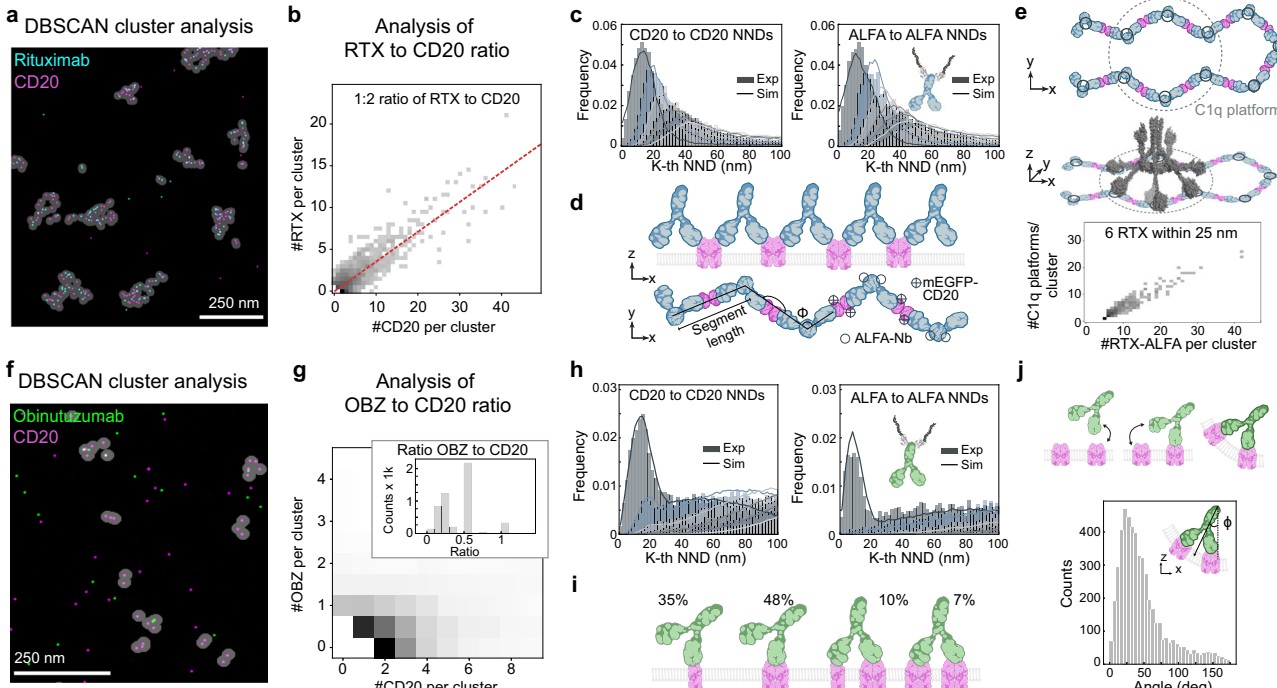

**Fig. 3 | Quantitative analysis and structural modeling of therapeutic antibody-CD20 complexes. a** DBSCAN analysis (marked in gray) of RTX (cyan)-CD20 (magenta) clusters shows distinct higher-order structures in 2D. **b** Quantitative analysis in 2D reveals a linear relationship between the number of RTX molecules and CD20 dimers. Two ALFA-Nbs correspond to one RTX per cluster. A linear fit yields 0.38, suggesting that approximately one RTX binds per CD20 dimer, when correcting for the labeling efficiency (for details, see Methods). **c** Nearest-Neighbor Distance (NND) analysis indicates a higher-order organization, which can be modeled by a flexible-chain arrangement. **d** The model includes anchor points at the RTX hinge regions, linear segments connecting these points, CD20 dimers located centrally along these segments and ALFA-Nbs located at the hinge regions. **e** The flexible-chain model can explain how RTX-CD20 interactions lead to the formation of U-shaped clusters to facilitate C1q binding. The number of hexameric RTX platforms was determined by counting each possible binding configuration of C1q (see Methods). **f** DBSCAN cluster analysis for OBZ (green) with CD20 in 2D

shows smaller clusters compared to RTX-CD20. **g** Quantitative analysis reveals specific OBZ to CD20 stoichiometries, without a linear relationship between the number of OBZ and CD20 molecules. Two ALFA-Nbs correspond to one OBZ per cluster. **h** NND analysis for CD20 complexes suggests that CD20 does not form higher-order structures beyond tetramers at the cell surface. NND analysis of the ALFA-Nbs labeling OBZ reveals only a first NND peak, representing two ALFA-Nbs bound per single OBZ. The absence of a second NND peak excludes a higher-order arrangement of OBZ-CD20 clusters. **i** Simulations with CD20 monomers, dimers, trimers and tetramers, taking into account the labeling efficiency of the GFP-Nb, for this representative cell result in 35 % monomers, 48 % dimers, 10 % trimers and 7 % tetramers after OBZ treatment. **j**, We observe a 25° angle between OBZ bound to CD20 and the xy-plane of the cell membrane. This suggests the necessity of membrane bending to allow for OBZ binding to CD20 with two Fab arms simultaneously. Created with the help of BioRender (https://BioRender.com/y76v9f4 and https://BioRender.com/5575210).

however, allowed us to faithfully resolve single proteins of both CD20 receptors and mAbs (Fig. 2b). When inspecting single clusters in 3D, it can be observed that RTX and CD20 organize in the membrane in a coplanar manner, with no apparent membrane bending induced by RTX binding (Fig. 2c, d and Supplementary Fig. 2). We detect RTX and CD20 with an average axial distance of 32 ± 11 nm, consistent with RTX bound outside of the cell membrane (Supplementary Fig. 3a).

In contrast, OBZ and CD20 co-cluster with a similar axial distance of 27 ± 9 nm (Supplementary Fig. 3b), but do not form >50 nm assemblies (Fig. 2e, f). RESI reveals the 3D oligomerization of CD20, forming a maximum of tetramers, when bound to OBZ (Fig. 2g, h). Excitingly, we were able to resolve two ALFA-Nbs bound to a single OBZ mAb, thus visualizing intramolecular distances within a protein (Fig. 2g). Interestingly, CD20 and OBZ do not seem to organize in a coplanar manner, suggesting potential membrane bending.

This detailed visualization with RESI highlights the different modes of interaction between CD20 and the therapeutic antibodies, RTX and OBZ, with implications for their mechanisms of action at the molecular level. To quantitatively analyze the properties of individual RTX-CD20 molecular assemblies, we first applied a cluster detection algorithm (Density-based spatial clustering of applications with noise, DBSCAN[20]) to the two-target RESI images (Fig. 3a). The stoichiometry of RTX:CD20, i.e., the number of RTX molecules per CD20 molecules, can be directly determined for each RTX-CD20 cluster, as RESI allows

us to resolve single protein copies of both receptors and mAbs. The number of CD20 molecules per cluster in RTX-treated cells ranges from a single CD20 molecule to over 40 molecules per cluster (Fig. 3b). Furthermore, Fig. 3b shows a clear linear correlation between the number of RTX molecules and the number of CD20 molecules in these clusters. When taking into account the labeling efficiency of ALFA- and GFP-Nbs (see Methods), a linear fit reveals approximately one RTX per two CD20 molecules (Fig. 3b, red line; Supplementary Fig. 4a), suggesting that CD20 dimers are linked by RTX molecules.

To further quantitatively evaluate the precise molecular arrangement of the RTX-CD20 assemblies, we analyzed the first to sixth nearest-neighbor distances (NND) for CD20 (Fig. 3c, histogram). Only with RESI, and not with DNA-PAINT we were able to routinely detect sub-10 nm distances, allowing us to recover the actual first-NND peak (Supplementary Fig. 5). To assess oligomeric CD20 and RTX arrangements, we then compared the NND histograms of the data to those of simulated point patterns (Fig. 3c, solid line; for simulation details, see the Methods section). RTX-treated CD20 shows higher-order arrangements for all NNDs when compared to complete spatial random (CSR) simulations (Supplementary Fig. 3c, d). Notably, CD20 NNDs after treatment with ALFA-tagged and untagged RTX are comparable, showing that the ALFA-tag does not affect the mAb's CD20 binding properties[16]. Consistently, our NND data aligns well with a flexible chain model, taking into account the highly flexible hinge

region of human IgG1 antibodies[21] (see Methods) (Fig. 3d and Supplementary Fig. 6a–c). We obtained an average chain segment length of 23±2 nm for the RTX hinge-to-hinge region distance (Supplementary Fig. 3e), which is longer than the expected length[22] of approx. 21 nm (see Methods). This is in line with the shallow binding angle for RTX-Fabs (Fig. 1b), leading to an increased Fab-to-Fab distance within the mAb[11,12].

We further assessed whether the flexible nature of RTX-CD20 chains can explain how RTX assemblies organize to form C1q binding platforms. C1q is a multimeric 460 kDa protein with six head domains, capable of activating the cytotoxic complement cascade upon binding to hexameric platforms of six mAbs[23].

We have previously shown that isolated hexameric circular platforms of CD20 and RTX, as postulated by Cryo-EM studies, are not compatible with our RESI data showing highly concatenated linear RTX-CD20 clusters[11,12,16]. Although circular arrangements may be present in limited amounts on the cell membrane, this suggests that an alternative structural organization leads to compatible C1q binding platforms.

For instance, U-shaped chains of ≥6 CD20 dimers could position ≥6 RTX-Fc domains in close proximity, allowing for efficient C1q binding (Fig. 3e, top). To test this hypothesis, we analyzed the number of C1q-compatible binding sites per RTX-cluster (see Methods) and indeed, we detected several platforms compatible with C1q binding (Fig. 3e, bottom; Supplementary Fig. 4b). Importantly, Fc-Fc interaction blockade does not change the structural RTX-CD20 arrangements observed with RESI, suggesting that this RTX-mediated CD20-clustering and the formation of C1q-binding platforms are independent of RTX Fc-Fc interactions (Supplementary Fig. 7).

In contrast to Type I RTX, analyzing Type II OBZ-CD20 co-clusters reveals no higher-order oligomerization and no linear relationship between the number of OBZ molecules and the number of CD20 molecules (Fig. 3f, g and Supplementary Fig. 4c). Instead, the frequencies of OBZ:CD20 stoichiometries result in discrete values, pointing towards limited oligomerization with one OBZ bound to one, two, three, or four CD20 molecules (Fig. 3g). To determine the oligomeric state of CD20, we first analyzed the 1st to 6th NNDs by comparing them with a simulated distribution of CD20 molecules according to CSR (Supplementary Fig. 3f and Supplementary Fig. 6d. For details on the CSR simulations, refer to Methods section "CD20 low order oligomerization simulations"). This comparison revealed specific peaks at distances below 20 nm for 1st to 3rd NNDs, which cannot be explained by a pure CSR distribution. These results indicate the presence of monomers, dimers, trimers, and tetramers of CD20 molecules on the cell surface (Fig. 3h). To assess the oligomeric state of the OBZ mAb when bound to CD20, we analyzed the NNDs of the ALFA-Nb used for OBZ-labeling (Fig. 3 h, bottom right). The ALFA-Nb to ALFA-Nb distances only show a non-random peak in the first NND, corresponding to the expected labeling with two ALFA-Nbs per single OBZ-molecule (Fig. 3h, right and Supplementary Fig. 3g). Moreover, the fact that the second NND peak is distributed according to CSR allows us to exclude the presence of higher-order assemblies of OBZ upon CD20 binding (Fig. 3h and Supplementary Fig. 6e), contrary to what we observed in the RTX case. Accordingly, we detected only a few compatible C1q binding platforms for OBZ (Supplementary Fig. 3h) as compared to RTX.

To quantitatively assess the experimental distribution of CD20 oligomers we compared our data with a simulated model of oligomers (monomers, dimers, trimers and tetramers) taking into account experimental factors such as labeling efficiency and linkage error (see Methods) (Fig. 3h). The proportions of each kind of oligomer are free parameters of the model that are retrieved from the fit to the data. We obtained a composition of 35% monomers, 48% dimers, 10% trimers and 7% tetramers of CD20 after OBZ binding (Fig. 3i and Supplementary Fig. 6c).

Contrary to the proposed terminal complex of CD20 tetramers[11], we detected fewer trimers and tetramers than expected after OBZ binding. Even though the detection efficiency of oligomers in RESI may be reduced due to two adjacent DNA-PAINT-unresolvable molecules labeled with the same DNA docking strand sequence, this cannot explain a trimer and tetramer proportion below 30%[16]. It could however be the case that the steep binding angle of OBZ-Fab (Fig. 1b, **bottom**) observed in Cryo-EM[11] necessitates membrane bending to allow for OBZ binding to two CD20 dimers simultaneously (Fig. 3j, top). To assess this experimentally, we measured the angle of the OBZ-CD20 complex, comprising CD20 trimers or tetramers, relative to the 2D plane of observation. Interestingly, we found an average angle of 25° (Fig. 3j, bottom and Supplementary Fig. 3i), suggesting local nanoscale bending of the cell membrane; however, we cannot exclude that tilting of the whole complex contributes to this phenomenon (Supplementary Fig. 4d). This energetically unfavorable membrane bending, which occurs when binding to two CD20 dimers happens simultaneously, likely accounts for the lower frequency of OBZ-induced CD20 tetramers.

## CD20 oligomeric arrangement determines Type II function

Intrigued by this finding, we next hypothesized that there is a potential to increase CD20 tetramerization by introducing a more flexible hinge region in the mAb. For this purpose, we focused on OBZ-based CD20-CD3-T cell engagers (TCEs)[24]. The TCEs were evaluated in two distinct formats. The classical format (c-TCE) resembles the anti-CD20 Fab arrangement of Type II OBZ, with an additional anti-CD3-Fab attached externally to one OBZ-Fab for T cell binding, while the inverted format (i-TCE) exchanges the positions of the anti-CD3-Fab and OBZ-Fab, placing the anti-CD3-Fab internally and the OBZ-Fab externally (Fig. 4a). After treatment with c- or i-TCE, we performed RESI imaging for CD20 like above. The RESI results show qualitatively comparable CD20 arrangements for both cases, with monomers, dimers, trimers and tetramers detected in the proximity of the mAbs (Fig. 4b, c and Supplementary Fig. 8). However, more detailed analysis using NND histograms yields a higher degree of oligomerization for i-TCE compared to c-TCE (Fig. 4d, e and Supplementary Fig. 9). By quantitatively comparing NNDs of the experimental data with NNDs of a numerical model simulating CD20 monomers, dimers, trimers and tetramers, we determine different trimer and tetramer proportions for both mAb formats, although the inter-CD20 dimer distance remains conserved (Fig. 4d, e and Supplementary Fig. 10). Strikingly, i-TCE is significantly more efficient in trimer and tetramer formation (49.6 ± 6.2%) compared to c-TCE (27.2 ± 10.5%), demonstrating that an increased linker flexibility through introducing the anti-CD3-Fab between the OBZ-Fabs indeed allows for more efficient CD20 tetramer formation (Fig. 4f).

To test whether this difference in CD20 trimerization and tetramerization capabilities for both TCE formats also results in a difference in CD20-directed function, we correlated the nanoscale molecular organization of CD20 with results obtained by functional assays in living cells. In a direct cell killing assay, assessing only the CD20-mediated cytotoxicity, (see Methods) only Type II, but not Type I mAbs potently induce direct cell death in CD20-positive cells (Fig. 4g and Supplementary Fig. 11). In agreement with its Type II functionality, treatment with c-TCE results in potent direct cell killing in CD20-positive cells (Fig. 4g and Supplementary Fig. 11). Conversely, the direct cell killing ability of i-TCE is significantly reduced, thus exhibiting more Type I-like functions (Fig. 4g and Supplementary Fig. 11). Moreover, i-TCE approaches Type I-like cell binding capabilities by engaging with about twice as many mAbs than c-TCE (Fig. 4h) in accordance with previously described Type I vs. Type II characteristics[25]. These two assays show that the function of c-TCE is comparable to that of Type II OBZ, whereas i-TCE exhibits a Type I-like function. Additionally, as we observed no higher-order oligomers beyond tetramers in the RESI images for either format, we conclude

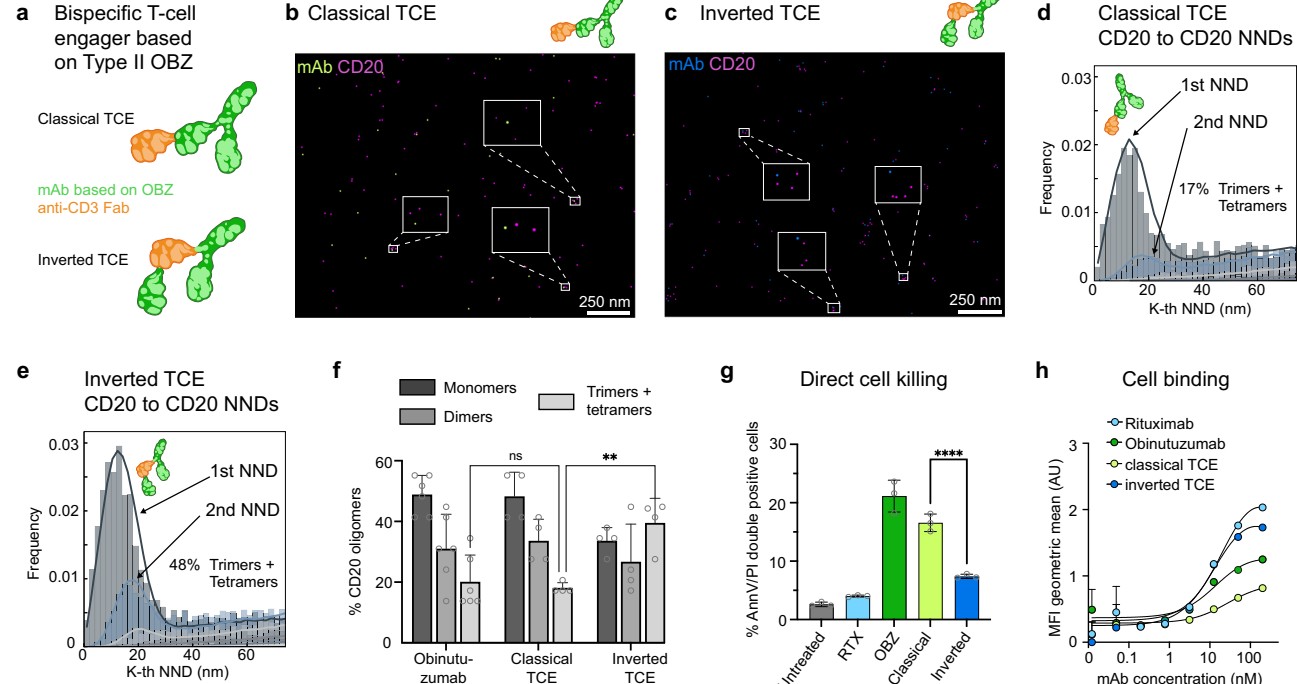

**Fig. 4 | Assessing the structure-function relationship of anti-CD20 antibodies.**
**a** Schematic representation of CD20-CD3 TCE configurations: classical (c-TCE) and inverted (i-TCE), featuring an anti-CD20 monoclonal antibody (mAb) based on OBZ and an anti-CD3 Fab fragment at two different positions in the molecule. **b**, **c** RESI images of mAb-CD20 clusters when treated with c-TCE (**b**) and i-TCE (**c**). **d**, **e** Nearest-Neighbor Distance (NND) analysis of CD20 clusters treated with c-TCE (**d**) and i-TCE (**e**), showing increased frequencies of non-random sub-25-nm distances in the second and third NND histograms for i-TCE. Least-squares fit (solid lines) of monomers, dimers, trimers and tetramers shows an increased trimer- and tetramerization for i-TCE compared to c-TCE. **f** Bar graph depicting the percentage of CD20 oligomers in monomeric, dimeric, and higher-order forms (trimers and tetramers) for OBZ, c-TCE, and i-TCE. The frequency of trimers and tetramers is increased for i-TCE compared to c-TCE. The bars and error bars represent the mean and standard deviation, respectively. The number of biological replicates is n(OBZ) = 3, n(c-TCE)=n(i-TCE) = 4. Statistical significance was tested using a two-way ANOVA, adjusting for multiple comparisons (p = 0.0027). Source data are

provided as a Source Data file. **g** Direct cell killing assay of CD20-positive Raji cells in untreated condition, and upon treatment with untagged versions of Rituximab (RTX), Obinutuzumab (OBZ), c-TCE, and i-TCE. i-TCE has a reduced killing efficiency compared to c-TCE, approaching values for Type I-RTX. The number of biological replicates is n = 3. The height of the bar and error bars represent mean and standard deviation, respectively. Statistical significance was tested using a one-way ANOVA, adjusting for multiple comparisons (p < 0.0001). Source data are provided as a Source Data file. **h** Graph depicting cell binding affinity in Raji cells across a range of mAb concentrations (0.01 to 100 nM) for untagged versions of RTX, OBZ, c-TCE, and i-TCE, measured as Mean Fluorescence Intensity (MFI) geometric mean. The results show an increased binding for i-TCE compared to c-TCE. The data points and error bars represent mean and standard deviation, respectively. The number of biological replicates is n = 3. Source data are provided as a Source Data file. Created with the help of BioRender (https://BioRender.com/y76v9f4 and https://BioRender.com/5575210).

---

that CD20 concatenation, as seen in the RTX case, is not required for the shift from Type II toward Type I functionality. Rather than higher-order oligomerization, we demonstrate that the efficient formation of CD20 trimers and tetramers induced by certain therapeutic mAbs is what drives the shift from Type II to Type I function.

### Therapeutic antibody function correlates with CD20 arrangement

To test if the CD20 oligomeric arrangement is a general determinant of Type I vs. Type II functionality, we next investigated CD20 oligomerization for additional Type I and Type II anti-CD20 mAbs. Type I Ofatumumab (OFA) leads to even stronger complement binding than RTX[26,27] and has also been shown to bind with two individual Fabs to CD20 dimers, albeit with a steeper binding angle[11,12]. Similar to RTX-treatment, RESI images acquired after OFA treatment show CD20 clustering (Fig. 5a and Supplementary Fig. 12a, b). In addition, the NND histograms reveal linear higher-order CD20 arrangements for all first to sixth NNDs, like RTX (Fig. 5b). Comparing the NND data to the same flexible chain-like model as for RTX allowed us to faithfully fit the CD20-NND data, albeit featuring a shorter (17 ± 2 nm) chain-segment length compared to RTX (23 ± 2 nm) (Fig. 5b and Supplementary Fig. 12c). This in fact agrees with the steeper binding angle of OFA-Fabs

to CD20 dimers, effectively reducing the intra-IgG1 Fab-Fab distance (Fig. 5c) while also forming platforms compatible with C1q binding (Supplementary Fig. 12d). Furthermore, we evaluated the Type I mAb 2H7, which shows higher-order oligomerization of up to hexamers (Supplementary Fig. 12e-g).

In addition to OBZ and c-TCE, we evaluated the CD20 oligomerization after treatment with the Type II clone H299. The RESI images of CD20 after H299 treatment reveal oligomers up to tetramers, comparable to OBZ data (Fig. 5d). NND histograms for H299 show a non-CSR first, second, and third NND peak, indicative of oligomers up to tetramers (Fig. 5e). These results are consistent with trimer and tetramer proportions that we previously detected for Type II mAbs (Fig. 4), yielding less than 30% of CD20 trimers and tetramers upon H299 binding (Fig. 5f).

Taken together, the unique ability of RESI to directly observe and quantify in situ molecular changes in CD20 arrangements allows us to propose a general model to correlate the mAb-induced CD20 oligomerization with their Type I or Type II function (Fig. 5g). Type II function, characterized by direct cell death, correlates with limited CD20 oligomerization of dimers, trimers and tetramers (Fig. 5g, left side). On the contrary, full Type I-like function, characterized by less efficient cell killing, is accompanied by the ability to form higher-order

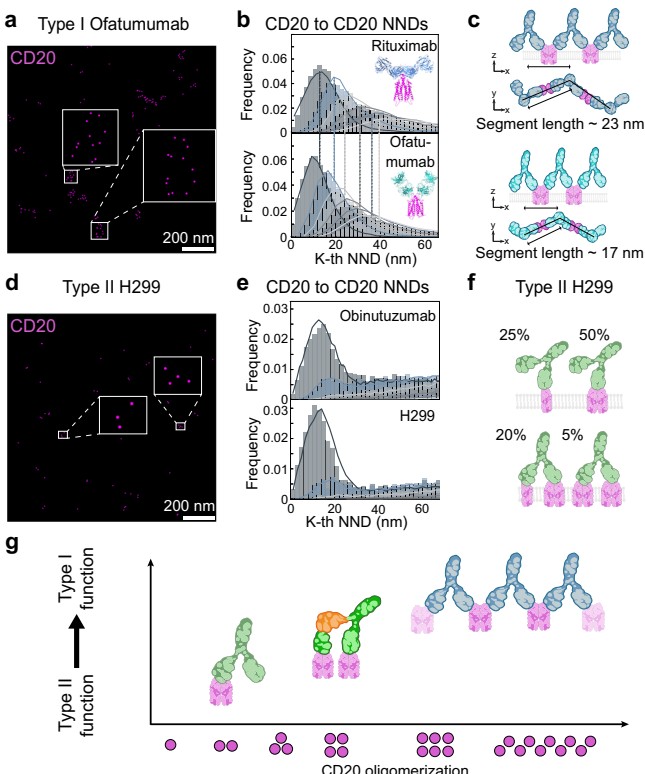

**Fig. 5 | 2-plex RESI imaging and quantitative analysis shows a structure-function relationship of Type I and Type II anti-CD20 therapeutic antibodies.**
**a** RESI image of CD20 treated with Type I Ofatumumab shows higher-order CD20 arrangements. **b** NND analysis (histogram) and fitting with CD20-flexible chain model (solid line). Image is representative of three independent experiments. **c** Type I Ofatumumab forms flexible chains with a shorter segment length than Rituximab, i.e. 17 ± 2 nm. **d** RESI image of CD20 treated with Type II H299 (clone used for therapeutic antibody Tositumomab) shows limited CD20 oligomerization. Image is representative of three independent experiments. **e** NND analysis (histogram) and fitting with a model of monomers, dimers, trimers and tetramers (solid line). **f** Type II H299 forms oligomers similarly to Obinutuzumab. **g** Correlation of CD20 Type I or Type II-like function with CD20 oligomerization. Created with the help of BioRender (https://BioRender.com/y76v9f4 and https://BioRender.com/5575210).

concatenated structures of at least hexamers, as shown by imaging of RTX, OFA, as well as 2H7 (Fig. 5g, right side). Notably, modifying the Type II-like c-TCE to the more flexible i-TCE, generates a transition from Type II-like to Type I-like function, as detected by a reduction of direct cytotoxicity. This is accompanied by a significant increase in the ability to form CD20 trimers and tetramers upon binding, thereby defining the molecular requirements for Type I and II function of therapeutic mAbs.

## Discussion

Our data highlights the importance of assessing the nanoscale organization of therapeutic monoclonal antibodies (mAb) and their target proteins during binding to gain a comprehensive understanding of the molecular mechanisms underlying their therapeutic action. We have developed a sensitive and versatile assay using RESI microscopy to quantitatively analyze the impact of mAbs binding to their cognate receptors on the nanoscale structural organization of receptor-mAb complexes within cells. RESI microscopy enables the imaging of individual molecules with Ångström-scale precision within intact cells, allowing for spatial analysis and modeling of the molecular functions of anti-CD20 mAbs. Our study reveals that Type I mAbs induce higher-order CD20 oligomerization with a minimum of hexamers, whereas Type II mAbs lead to limited CD20 oligomerization, forming up to tetramers. We evaluated the effect of changing the relative orientation of the two OBZ-Fab arms by inverting the anti-CD3 and anti-CD20 Fab arms in the OBZ-like CD20-CD3-TCE. When using CD20-CD3 TCEs in the presence of T cells, i-TCE is a more potent bispecific T cell engager than c-TCE, most likely due to the highly efficient CD20 binding and closer spatial contact of cancer cells with T cells[24]. Interestingly, when

only investigating the CD20-mediated direct cytotoxicity, independent of T cell-mediated effects, we found that i-TCE reduces the original direct cytotoxicity of c-TCE while it increases the formation of CD20 trimers and tetramers. This suggests that a higher abundance of CD20 trimers and tetramers inversely correlates with the direct cytotoxicity of anti-CD20 antibodies. This partial loss of direct cytotoxicity while also showing a partial increase in CD20 oligomer formation in the case of i-TCE suggests a continuum in both therapeutic function and CD20 oligomerization between Type I and Type II mAbs, rather than two distinct categories.

According to Type I-like gain of function with increasing CD20 oligomerization, Type II OBZ has previously been shown to mainly activate cell death inducing pathways, while Type I RTX has a dual function in activating both pro-apoptotic and anti-apoptotic signaling pathways within the B-cell receptor (BCR) cascade[28,29]. This suggests that increasing CD20 oligomerization by Type I mAbs promotes pro-survival effects induced by CD20-mediated signaling. The extent to which therapeutic mAb-modulated CD20 oligomerization influences BCR signaling cascades needs further investigation, as this has important implications on personalized medicine or combination therapy with drugs influencing the BCR cascade[30].

Only with RESI, we were able to image the macromolecular assembly of CD20 and mAbs in the cellular context, revealing that Type I mAbs can form platforms compatible with C1q binding within chain-like arrangements, without the need for closed rings (Figs. 2 and 3)[31]. Shorter antigen-to-antigen distances correlate with C1q deposition and activation[32], which is in agreement with the shorter chain segment length we deduced for OFA vs. RTX, as OFA is known to lead to more stable C1q binding[27]. Furthermore, the flexible chain

model explains how adjacent C1q platforms within a chain could allow for C1-complex (C1q with proteases C1r and C1s bound) cross-activation[33,34].

Our findings demonstrate the universal applicability of our RESI imaging and analysis workflow for any membrane protein that holds potential as an immunotherapeutic target. We can elucidate oligomeric as well as nano- to microscale structural arrangements of membrane proteins in complex with their cognate antibodies in situ, i.e. in whole intact cells, a capability thus far out of reach for super-resolution microscopy. RESI features a dynamic range spanning almost five orders of magnitude, from whole cells (10-100 µm) to intermolecular distances of protein dimers (1-10 nm).

MINFLUX and related techniques achieve localization precisions of 1–2 nm (sub-5 nm resolution), comparable to RESI and sufficient to resolve single proteins[35–39]. When combined with DNA-PAINT, MINFLUX also allows for high multiplexing[40]. However, throughput is currently a major limitation in MINFLUX: areas of a few hundred nanometers are acquired sequentially (several minutes of acquisition each), thus requiring hours for fields-of-view spanning only a few micrometers[35,40]. While whole-cell imaging with MINFLUX is theoretically possible, it has not yet been demonstrated. In contrast, RESI can acquire ~100 × 100 µm² fields-of-view—covering multiple cells—in similar timeframes, yielding a ~1000-fold higher throughput.

The current throughput of RESI enables imaging of more than 10 cells per day, making it a powerful tool for screening therapeutic mAb candidates at molecular resolution. Moreover, automation and parallelized measurements in multiple microscopes could extend the throughput beyond hundreds of cells per day. Training machine learning models on RESI data could enable the prediction of mAb functions based on their oligomeric patterns, offering potential for biosimilar and generic drug screening, as well as quality control. To further refine the relationship between the structure and function of anti-CD20 mAbs, future studies could investigate those mAbs that have been shown to unify Type I and Type II functionalities[41].

RESI microscopy will not only enhance our understanding of existing therapies but also pave the way for the rational design of next generation mAbs with optimized therapeutic profiles. Future research could focus on the relationship between mAb structure and function across different cancer types and treatment contexts to fully exploit the potential of these biological agents.

## Methods

### Materials
Right-handed DNA oligonucleotides, modified with C3-azide or 5′-Cy3B, were ordered from Metabion. Left-handed DNA sequences were purchased from Biomers. Ultrapure water (cat: 10977-035), Tris 1 M pH 8 (cat: AM9855G), EDTA 0.5 M pH 8.0 (cat: AM9260G) and 10×PBS pH 7.4 (cat: 70011051) were purchased from Thermo Fisher Scientific. Sodium chloride 5 M (cat: AM9759), Bovine serum albumin (cat: A9647) and sodium azide (cat: 71289 were ordered from Sigma-Aldrich. Lipofectamine LTX (A12621) and 16% Formaldehyde methanol-free (cat: 28908) and sheared salmon sperm DNA (cat: AM9680) were obtained from Thermo Fisher Scientific. Glutaraldehyde (25%, cat: 4157.1) and NH₄Cl (cat: K298.1) were ordered from Carl Roth. Triton X-100 (10% solution) (cat: 93443), Tween 20 (cat: P9416-50ML), glycerol (cat: 65516-500 ML), protocatechuate 3,4-dioxygenase pseudomonas (PCD) (cat: P8279), 3,4-dihydroxybenzoic acid (PCA) (cat: 37580-25G-F) and (+−)-6-hydroxy-2,5,7,8- tetra-methylchromane-2-carboxylic acid (Trolox) (cat: 238813-5 G) were ordered from Sigma-Aldrich. Fetal Bovine Serum (FBS) (cat. A5669701, Gibco), 1× Phosphate Buffered Saline (PBS) pH 7.2 (cat: 20012-019), 0.05% Trypsin−EDTA (cat: 25300-054), Lipofectamine 3000 (cat: L3000015) were purchased from Thermo Fisher Scientific. 90 nm diameter Gold Nanoparticles (cat: G-90-100) were ordered from Cytodiagnostics. µ-Slide 8 Well high Glass Bottom (cat: 80807) was purchased from ibidi.

Amicon Ultra-0.5 and Amicon Ultra-2 centrifugal filter units with 10k and 50k MWCO (cat: UFC5010, UFC 5050, UFC201024, UFC205024) were purchased from Merck.

### Cloning
mEGFP-CD20 was cloned by insertion of CD20 into the mEGFP-C1 plasmid (no. 54759, Addgene). A CD20 gblock (obtained from Integrated DNA Technologies) was inserted with Gibson assembly after cutting with restriction enzymes BsrGI and BamHI (2× Gibson Assembly mix, New England Biolabs cat: E2611).

### Nanobody-DNA conjugation via a single cysteine
Nanobodies against GFP, ALFA, and human IgG were ordered with a single ectopic cysteine at the C-terminus for site-specific and quantitative conjugation. The conjugation to DNA-PAINT docking sites (see Supplementary Tables 1 and 2) was performed as described previously[16]. First, buffer was exchanged to 1× PBS + 5 mM EDTA, pH 7.0 using Amicon centrifugal filters (10k MWCO) and free cysteines were reacted with 20-fold molar excess of bifunctional maleimide-PEG₄-DBCO linker (Sigma-Aldrich, cat: 760668) for 2-3 hours on ice. Unreacted linker was removed by buffer exchange to PBS using Amicon centrifugal filters. Azide-functionalized DNA was added with 3-5 molar excess to the DBCO-nanobody and reacted overnight at 4 °C. Unconjugated nanobody and free azide-DNA were removed by anion exchange chromatography using an ÄKTA Pure liquid chromatography system equipped with a Resource Q 1 ml column. Nanobody-DNA concentrations were adjusted to 5-10 µM (in 1xPBS, 50% glycerol, 0.05% NaN₃) and stored at −20 °C for 1-6 months or at -80 °C for >6 months.

### Cell culture
CHO-K1 cells (CCL-61, ATCC) were cultured in Gibco Ham's F-12K (Kaighn's) medium (Thermo Fisher Scientific, cat: 21127030), supplemented with 10% FBS (cat. A5669701, Gibco). Cells were passaged every 2–3 days using trypsin-EDTA.

### ALFA-tagged therapeutic antibodies
The ALFA-tag was fused genetically to the C-terminus of the heavy chain of rituximab and obinutuzumab via a G5S linker (GGGGGSPSRLEEELRRRLTE). The respective antibodies were transiently expressed in HEK239 cells and purified via protein A. Their identity and activity were confirmed (Supplementary Table 3).

### Expression and purification of SpA-B
A plasmid (pET-30b(+)_SpA-B) enabling bacterial expression of *Staphylococcus aureus* Protein A, subunit B (SpA-B), was constructed by Gibson Assembly, inserting the codon-optimized gene (Uniprot entry P38507, A212-K269) into a pET-30b(+) (Novagen, cat: 69910) plasmid previously digested with NdeI and XhoI endonucleases. The resulting construct included a C-terminal Sortase A (LPETGG) sequence, followed by a hexahistidine tag to facilitate subsequent purification and functionalization.

An aliquot of *Escherichia coli* BL21(DE3) (New England Biolabs, cat: C2527H) was transformed with pET-30b(+)_SpA-B and a single colony was picked and inoculated into 1 L of LB_KAN medium, followed by incubation at 37 °C, 200 RPM. When the OD600 reached 0.5, protein expression was induced by adding IPTG to a final concentration of 0.2 mM, and incubation was continued at 37 °C, 200 RPM for 4 hours. Cells were harvested by centrifugation (10,000 × g, 15 minutes), resuspended in 40 mL of PBS, and lysed by sonication (3 minutes, 40% amplitude, 1 s on/off pulses) on ice. The lysate was centrifuged (15,000 × g, 30 minutes), filtered, and the supernatant was loaded onto a Ni-NTA (HisTrap FastFlow 5 ml, Cytiva, cat: 17528601) column pre-equilibrated with PBS. The column was washed with 20 mM imidazole in PBS, and the protein was eluted with 250 mM imidazole in PBS.

The eluate was concentrated using a 3 kDa cut-off filter and loaded onto a Superdex 75 Increase 10/300 GL column (Cytiva, cat: 17517401), with PBS as the mobile phase. The protein eluted as a single peak, and the corresponding fractions were pooled (5.8 mg/mL), aliquoted, frozen in liquid nitrogen, and stored at −20 °C

## Binding of SpA-B to IgG1

Specific binding of SpA-B to IgG1 was assessed by first immobilizing 3 mg SpA-B in His-Incubation buffer (Supplementary Table 4) on Ni-NTA beads (Ni-NTA Spin Purification Kit, Thermo Scientific, 88227) for 30 min at RT. Then, Ni-NTA beads were washed 4 times with His-Washing buffer (Supplementary Table 4). The SpA-B loaded beads were incubated with RTX-IgG1 in His-Incubation buffer for 30 min at RT. After incubation, SpA-B and RTX-IgG1 coated beads were again washed 4 times with His-Washing buffer before elution with His-Elution buffer (Supplementary Table 4). To control for unspecific binding of both SpA-B and RTX-IgG1 to Ni-NTA beads, 3 mg BSA instead of SpA-B were incubated on Ni-NTA beads, followed by washing as above. The BSA coated beads were subsequently incubated with RTX-IgG1, followed by washing and elution as above. Samples were analyzed on an SDS-PAGE.

## Treatment with therapeutic antibodies

CHO-K1 cells were seeded on 8 Well high Glass Bottom chambers the day prior to transfection at a density of 10,000 cells per well. CHO cells were transfected with mEGFP-CD20 using Lipofectamine LTX as specified by the manufacturer. CHO cells were allowed to express receptors overnight. Therapeutic antibodies were thawed on ice, subjected to a spin at >20000xg at 4 °C for 10 min and the supernatant was kept on ice until cell treatment. F12K medium + 10% FBS was heated to 37 °C. All therapeutic antibodies (RTX-Alfa and OBZ-Alfa, OFA, 2H7, CD20-CD3-TCEs) were diluted to 66.7 nM in medium (Supplementary Table 3). For the experiments assessing the effect of Fc-Fc interaction blockade, SpA-B was incubated with RTX or OBZ at a 20-fold molar excess for 15 min at RT, before adding the mAbs to the pre-heated medium. Cells were incubated with therapeutic antibodies for 30 min at 37 °C. Next, cells were washed 2x with 250 μL F12K medium. 16% PFA was diluted 1:4 in 1xPBS and pre-heated to 37 °C for 10 min. The medium was removed, and cells were immediately fixed with 250 μL 4% PFA for 15 minutes and washed 3x with PBS. Cells were permeabilized in 0.1% TritonX-100 in PBS for 5 minutes and washed with PBS followed by incubation with Blocking buffer (Supplementary Table 4) for 1 h at RT.

## RESI sample preparation

The RESI staining mix was prepared as follows: anti-GFP Nbs (clone 1H1), conjugated with DNA-docking strands 5xR1, 5xR2, 7xR3, or 7xR4 were added in equimolar amounts to a final concentration of 50 nM in Blocking buffer, and ALFA-tag nanobodies conjugated with DNA-docking strands 5xR5, 5xR6, 7xL1 or 7xL4 were added to the GFP-Nb-mix in Blocking buffer to a final concentration of 50 nM. 200 μL RESI staining mix was added per well and incubated overnight at 4 °C. The next day, unbound binders were removed by washing with the Washing buffer (Supplementary Table 4), followed by a single wash with PBS. Post-fixation was performed with 4% PFA in PBS + 0.1% glutaraldehyde for 10 min. Quenching was performed with 0.2 M NH₄Cl in PBS, followed by washing with PBS. Gold nanoparticles, freshly diluted 1:3 in PBS, were vortexed thoroughly, and 250 μL of the gold suspension was added and incubated for 7 min before washing with PBS.

## RESI imaging

mEGFP-CD20 transfected cells were selected by screening for homogenous, low GFP-fluorescence in TIRF mode, using a 488 nm excitation laser with 1 mW at the objective. Next, the Imaging buffer (Supplementary Table 4) was freshly prepared, and R4 imager strands were added at a concentration of 500 pM. Using the 561 nm excitation laser

at 30 mW at the objective, cells were further selected for sufficiently sparse, homogenous blinking density. Afterwards, the first RESI image acquisition round was initiated with imager concentration ranging from 500 pM to 1 nM. Imagers were washed off with PBS, and imager strands for the next RESI imaging round were added in a freshly prepared Imaging buffer (Supplementary Table 4). All other imagers were added with the same procedure until 8 rounds of RESI imaging were completed. A detailed overview of imaging parameters is listed in Supplementary Table 5.

## Microscope setup

2D and 3D Fluorescence imaging was carried out on an inverted microscope (Nikon Instruments, Eclipse Ti2) with the Perfect Focus System, applying an objective-type TIRF configuration equipped with an oil-immersion objective (Nikon Instruments, Apo SR TIRF ×100, numerical aperture 1.49, oil). 488 nm and 560 nm lasers (MPB Communications, 1 W) were used for excitation and coupled into the microscope via a Nikon manual TIRF module. The laser beams were passed through cleanup filters (Chroma Technology, ZET488/10× for 488 nm excitation, ZET561/10× for 560 nm excitation) and coupled into the microscope objective using a beam splitter (Chroma Technology, ZT488rdc-UF2 for 488 nm excitation, ZT561rdc-UF2 for 560 nm excitation). Fluorescence was spectrally filtered with an emission filter (Chroma Technology, ET525/50 m and ET500lp for 488 nm excitation, ET600/50 m and ET575lp for 560 nm excitation and ET705/72 m) and imaged on an sCMOS camera (Hamamatsu, ORCA-Fusion BT) without further magnification, resulting in an effective pixel size of 130 nm (after 2 × 2 binning). The central 1,152 × 1,152 pixels (576 × 576 after binning) of the camera were used as the region of interest, and the scan mode was set to lowest noise (≥100 ms) for 100 ms integration time and low noise (≥25 ms) for 75 ms integration time. Three-dimensional (3D) imaging was performed using an astigmatism lens (Nikon Instruments, N-STORM) in the detection path[42]. Raw microscopy data were acquired using μManager (Version 2.0.1)[43].

## Image analysis for RESI

Raw fluorescence data were subjected to super-resolution reconstruction using the Picasso software package[7] (latest version available at https://github.com/jungmannlab/picasso). Drift correction was performed with redundant cross-correlation and gold particles as fiducials for cellular experiments. Gold NPs were also used to align all rounds for 8-plex exchange. After channel alignment with RCC and gold NPs, whole cell regions of the DNA-PAINT data were picked and analyzed using the Picasso SMLM clustering algorithm (latest version available at https://github.com/jungmannlab/picasso) for each target individually.

## Clustering of DNA-PAINT localizations

Cluster analysis was performed according to a previously published method[16]. After channel alignment, DNA-PAINT data were analyzed using a custom clustering algorithm in 'Picasso: Render'. First, localization cloud centers were identified via gradient ascent. Localizations surrounding each center could be grouped, due to sufficient spacing between localization clouds in RESI.

The algorithm requires two input parameters: radius r, defining the cluster size and circular area around each localization, and $n_{min}$, the minimum number of localizations per cluster. The number of neighbors within distance r of each localization is counted. If a localization has more neighbors than its surrounding points, it is marked as a local maximum. Clusters are formed if there are more than $n_{min}$ localizations within r of a local maximum; non-clustered localizations are discarded. RESI localizations are the centers of the localization groups, calculated as weighted mean by employing the squared inverse localization precisions as weights.

Clusters are filtered in two ways to exclude those of non-specific "sticking" of imagers. First, the mean frame of localizations in each cluster is calculated, and clusters with a mean frame in the first or last 20% of frames are excluded. Second, clusters containing over 80% of localizations within 5% of the frames are also excluded.

The clustering radius (r) and threshold ($n_{min}$) depend on experimental conditions. A suitable $n_{min}$ can be determined by selecting localization clouds from single target molecules, plotting a histogram, and distinguishing between targets and background. Radius r should be approximately $1.5 \times$ to $2 \times$ the NeNa localization precision to avoid overlapping clusters or sub-clustering. Picasso Render provides a tool to test different clustering parameters on small regions of interest.

For 3D clustering, the xy-clustering radius r is doubled to be used as the z radius, as the z the spread of localizations in z is approximately two-fold greater compared with x and y.

### 3D visualization
Regions of interest, i.e. single clusters for RTX/CD20 or OBZ/CD20 were picked from the 2-target image and individually displayed as a 2-target image in the 3D window in Picasso Render. For visualization in 3D, the individual picks were rotated by 90° or 270° around the x- or the z-axis. 3D DNA-PAINT data was displayed as clustered localizations or as RESI localizations.

### DBSCAN two-target cluster analysis of RESI data
2-target RESI images of RTX/CD20 and OBZ/CD20 were subjected to DBSCAN[20] co-clustering with a radius of 40 nm and a minimum of at least 1 Alfa-Nb and 1 GFP-Nb detected. Non-clustered localizations were discarded for 2-target analysis.

### Ratio analysis
After DBSCAN co-clustering, ALFA-Nb to GFP-Nb ratios were determined on a per cluster basis and the histogram was plotted. Moreover, counts of ALFA-Nbs per cluster were plotted versus GFP-Nb counts and represented in a 2D histogram. To determine the correlation, a linear fit was performed with Python polyfit and the slope of the linear fit was determined. The slope was normalized for Alfa-Nb and GFP-Nb labeling efficiencies (40% and 50%, respectively) to determine the mAb to CD20 ratio[44]. Graphs in Fig. 3 were normalized so that 2 ALFA-Nbs equals 1 therapeutic antibody (RTX or OBZ).

### C1q platform analysis
C1q is a complex with 6 head domains, each capable of binding to one antibody-Fc domain and a hexameric platform of Fc domains is the most efficient[32] in capturing C1q. Consequently, 6 therapeutic antibodies in spatial proximity were approximated as ≥ 5 ALFA-Nbs within 25 nm, as ALFA-Nbs were previously measured to label with 40% labeling efficiency[44]. Importantly, C1q-binding platforms are defined as one possible configuration for binding, without removing clusters containing the same ALFA-Nb, to represent the apparent gain in avidity.

### Angle analysis
For each OBZ-CD20 co-cluster containing more than 3 CD20 molecules, the center of mass for both the mAb and the CD20 cluster was determined. Then, the vector from mAb-center of mass to CD20-center of mass was calculated and the angle between this vector and the normal vector on the xy-plane (which corresponds to the plane of a flat membrane on the coverslip) was determined (see Fig. 3h). Therefore, an angle of 0° means a perfectly perpendicular mAb over the CD20 cluster.

### Distance analysis
The vector from mAb-center of mass to CD20-center of mass was determined for each cluster, as described in angle analysis and the absolute value of the vector was determined (see Fig. 3h).

### DBSCAN one-target cluster analysis
CD20 RESI data was subjected to DBSCAN clustering with a radius of 20 nm and a minimum of at least 2 GFP-Nb detected per cluster. Individual clusters were identified, and non-clustered molecules were discarded.

### CD20 chain-like oligomerization simulations
CD20 cluster size frequencies and the density of CD20 clusters were extracted from the DBSCAN one-target cluster analysis data. Then, a random set of x-y coordinates was generated as chain-starting points. For each individual chain, the number of chain segments was randomly selected according to the experimentally evaluated cluster size frequency. Then, for each chain-starting point, a random angle was chosen to simulate the next hinge point. From this hinge point, an angle between 30 and 330 was randomly chosen to simulate the next hinge point. This process was repeated for each chain, until the maximum number of chain-segments was simulated. Each center of mass between two hinge points was used to simulate 2 GFP-Nbs labeling a CD20 dimer. The hinge points were used to simulate two ALFA-Nbs labeling the therapeutic antibody. The resulting GFP-CD20 and Alfa-Nb coordinates were subjected to labeling efficiency and uncertainty corrections and the nearest-neighbor distance histograms were analyzed. Fixed parameters for simulations were experimentally determined chain-lengths and chain-frequencies, the hinge angles, as estimated to be between 30° and 330° from the domain plane angle analysis of Individual Particle Electron Tomography data[21], the experimentally determined 13.5 nm GFP-Nb to GFP-Nb dimer distance[16] and the experimentally determined ALFA-Nb to ALFA-Nb distances of 4 nm. Segment length was a free parameter (i.e. hinge point to hinge point distance) and was optimized starting from an estimated distance of <21 nm, considering a ~16 nm Fab-to-Fab distance[22] plus contributions of the ~5 nm CD20 dimer distance[11,12].

### Monte-Carlo simulations to determine the uncertainty of the chain segment length
Three separate "optimal" CD20 chain simulations with the optimal chain segment length (RTX: 23 nm, OFA: 17 nm) were generated from the experimental data. Each of the simulated data sets were subjected to cluster analysis to obtain cluster sizes. According to the cluster sizes of these "optimal" simulations, chain simulations iterating through different chain segment lengths, were performed. The 1st to 10th NNDs of "optimal" and "iterating" simulations were determined. Then, the sum of least squares between the "iterating" NNDs and the respective "optimal" simulation were calculated. The results are displayed in the respective Extended Data Figures. Significant differences between the sum of least squares were assessed with an unpaired t-test. The first significant difference in the chain segment lengths was defined as the uncertainty of the deduced chain segment length.

### CD20 low order oligomerization simulations
Expected coordinates of CD20 oligomers, such as monomers, dimers, trimers, tetramers were generated. Simulations of variable proportions of CD20 oligomers were performed. For the data and for each simulation, NND analysis was performed. The most likely proportions of populations of oligomers were obtained through a least-squares optimization procedure. Simulation and analysis were performed with the Picasso module SPINNA[45].

The algorithm for optimizing oligomer proportions of the simulation can be summarized as follows:
- <u>Simulation of monomers</u>: a set of spatial coordinates with CSR distribution and given density are drawn.
- <u>Simulation of dimers</u>: a set of spatial coordinates with CSR distribution are drawn, representing the center of each dimer. For

each center, two positions are generated with a random orientation and expected distances.

- Simulation of trimers: a set of spatial coordinates with CSR distribution are drawn, representing the center of each trimer. For each center, three positions are generated with a random orientation and expected distances.
- Simulation of tetramers: a set of spatial coordinates with CSR distribution are drawn, representing the center of each tetramer. For each center, four positions are generated with a random orientation and expected distances.
- The final position of each molecule is computed taking into account the *Uncertainty* parameter (in all cases 5 nm) drawn from a gaussian distribution.
- NNDs are calculated on the subset of detectable molecules, corrected by the published labeling efficiencies[44].
- The number of dimers, trimers and tetramers per unit area are simulated such that the *total density* of molecules is set to match the observed experimental density after taking into account the labeling efficiency of the molecules.

## Cell Culture for cell binding and direct cell killing assays
Raji (ECACC #85011429) were cultivated in DMEM supplemented with 10% FCS and 1% Glutamax (Invitrogen/Gibco # 35050-038), SU-DHL-6 (ATCC #CRL-2959) and OCI-LY18 (DSMZ #ACC699) were cultivated in RPMI1640 supplemented with 10% FCS and 1% Glutamax.

## CD20 binding assay
CD20 binding of obinutuzumab, rituximab, glofitamab and CD10 classical TCB was assessed on OCILY18, SUDHL6 and Raji B cell lines. The cell lines were resuspended at $1 \times 10^6$/ml in FACS Buffer and 100 μl/well (100000/well) were seeded into 96-U-bottom plates. Antibody dilutions were prepared in FACS Buffer (200 nM down to 0.01 nM, 1:4 dilution steps). 25 μl/well of the pre-diluted antibodies or PBS were added after centrifugation to the cell pellets and incubated for 1 h at 4 °C. Afterwards, cells were washed and incubated with a PE-labelled secondary antibody (PE-conjugated AffiniPure F(ab)'₂ Fragment goat anti-human IgG Fcg Fragment specific (Jackson Immunoresearch Lab 109-116-170) in the presence of a live/dead marker (NIR) for 30 min at 4 °C. Cells were washed twice and re-suspended in 150 μl/well FACS Buffer/PBS and measured using a BD FACS CantoII.

## Assessment of direct cell death
Phosphatidylserine exposure and cell death were assessed by FACS analysis of Annexin V- (Annexin V FLUOS Staining Kit, Roche Applied Science #11828681001) and PI (Sigma Aldrich #P4864)-stained cells. Briefly, $1 \times 10^5$ target cells/well (190 μL/well) were seeded in 96-well plates and incubated with mAb (12.5 nM) for 24 h (untreated samples were used as negative control). Cells were then washed with Annexin V binding buffer (10 mM HEPES/NAOH pH7.4, 140 mM NaCl, 2.5 mM CaCl₂), stained with Annexin V FITC for 15 min at room temperature in the dark, then washed again and re-suspended in Annexin V binding buffer (200 μL/well) containing PI. Samples were analyzed immediately on a BD FACSCantoTM II.

## Reporting summary
Further information on research design is available in the Nature Portfolio Reporting Summary linked to this article.

## Data availability
All data are included in the Supplementary Information or available from the authors, as are unique reagents used in this Article. The raw numbers for charts and graphs are available in the Source Data file whenever possible. Localization data are available on Zenodo (https://doi.org/10.5281/zenodo.15552356). Source data are provided with this paper.

## Code availability
Picasso can be downloaded from GitHub: https://github.com/jungmannlab/picasso. Oligomer analysis was performed with the Picasso module SPINNA[45].

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

## Acknowledgements

We thank Cindy Schulenburg for expressing ALFA-tagged versions of Rituximab and Obinutuzumab. Therapeutic antibodies were provided by Roche Glycart. This research was funded in part by the European Research Council through an ERC Consolidator Grant (ReceptorPAINT, Grant agreement number 101003275, R.J.), the BMBF (Project IMAGINE, FKZ: 13N15990, R.J.), the Volkswagen Foundation through the initiative 'Life?—A Fresh Scientific Approach to the Basic Principles of Life' (grant no. 98198, R.J.), the Max Planck Foundation and the Max Planck Society. IP and SCMR acknowledge support by the IMPRS-ML graduate school. LAM acknowledges a postdoctoral fellowship from the European Union's Horizon 20212022 research and innovation program under Marie Skłodowska-Curie grant agreement no. 101065980. O.S. acknowledges support from the Czech Science Foundation grant no. 24-12553 O. Molecular graphics were generated with UCSF ChimeraX, developed by the Resource for Biocomputing, Visualization and Informatics at the University of California, San Francisco, with support from National Institutes of Health R01-GM129325 and the Office of Cyber Infrastructure and Computational Biology, National Institute of Allergy and Infectious Diseases. Part of the schematics were generated with the help of BioR-ender (www.biorender.com).

## Author contributions

I.P. conceived and performed all experiments except the functional studies, designed analysis software and analyzed the data. L.A.M. designed and wrote the analysis software and analyzed the data. S.C.M.R. wrote analysis software. J.K. produced DNA-conjugated label-ing reagents. O.S. expressed and purified the SpA-B protein. M.L., S.H. and M.B. designed and performed functional studies. C.K. contributed to the design of studies targeting CD20 and interpretation. I.P., L.A.M. and R.J. interpreted data and wrote the manuscript. R.J. conceived and supervised the study. All authors reviewed and approved the final manuscript.

## Funding

## Competing interests

M.L., S.H., M.B. and C.K. declare employment, patents (unrelated to this work) and stock ownership with Roche. The other authors declare no competing interests.

## Additional information

**Peer review information** *Nature Communications* thanks Alistair Curd, Marcin Okrój and the other, anonymous, reviewer(s) for their contribu-tion to the peer review of this work. A peer review file is available.

