## [Transparent Peer Review file · Nature Communications]

Resolving the structural basis of therapeutic antibody function in cancer immunotherapy with RESI

Corresponding Author: Professor Ralf Jungmann

Version 0:

Reviewer comments:

Reviewer #1

(Remarks to the Author)

The manuscript of Pachmayr et al. extends the scope of resolution enhancement by sequential imaging (RESI) to multiple targets and the third dimension and uses this technique to characterize and differentiate Type I and Type II mAbs targeting CD20. RESI's capability to localize individual CD20 molecules in cell systems with sub-nanometer resolution is impressively shown and forms the basis of subsequent discussions of CD20 clustering induced by various mAbs, the colocalization of these molecules, and structural characteristics of the resulting complexes. The inclusion of modified antibodies and functional data, leading to a mechanistic insight of the relationship between CD20 oligomerization and Type I vs Type II mAbs makes this a consistent and well-rounded manuscript.

A 2023 hallmark study by the same group introduced RESI and already used one of the CD20 type I mAbs employed in this manuscript, Rituximab (RTX). Notably, the previous study used unmodified RTX while the present manuscript added nanobodies and RESI sequences to the RTX Fc. This notwithstanding, both studies essentially produce the same NND distribution of CD20 after RTX treatment. This point should be highlighted in the manuscript, as it shows that the employed labeling strategy does not change RTX-induced CD20 clustering, as might be expected by the substantial RESI labels used.

The authors demonstrate that Type I mAbs form structures termed "C1q binding platforms", while Type II mAbs can do so only to a severely reduced extent, attributing this platform formation to large U-shaped clusters. While the presented quantitative cluster analysis and NND data supports this claim, this reviewer strongly suggests performing additional experiments to gauge the role of Fc-Fc interactions in the formation of these clusters. IgG oligomerization is the accepted mechanism of specific C1q recruitment and C1 activation leading to CDC, and IgG hexamerization would be consistent with the presented number of RTX per cluster (Fig 3b), and thus should be adequately considered and discussed. This may be achieved by using CDC-enhanced RTX and OBZ point mutants that have been shown to increase IgG oligomerization via Fc-Fc interactions (e.g. E345R, E345K, or E430G) and gauging the effects on observed cluster sizes and NNDs, as was already done for c-TCE and i-TCE. IgG oligomerization inhibiting mutations may also be used to rule out significant contributions of this process to the CD20 clustering observed by RTX. Alternatively, Fc blocking agents such as a Fc binding peptide like DCAWHLGELVWCT used by Diebolder et al. (Science. 2014 March 14; 343(6176): 1260–1263) or derivatives of Staphylococcal protein A (PNAS. 2021 February 9; 118 (7) e2016772118) might be employed to reversibly block IgG oligomerization in this case.

Showing actual C1q binding data to correlate with C1q platform abundances would further improve this point.

Finally, were the direct cell killing assays performed with the same labeled mAbs as the rest of the study? This point was not clear to this reviewer but may be critical to interpreting these results.

Specific comments:

1) Figure 1 shows models of antibody-mediated CD20 arrangements (b) and actual RESI localizations (c) side by side. The model uses one colored dot to represent a monomer, while in the real data two localizations are recorded per antibody (but not CD20) as per the labeling strategy. Consider standardizing the number of signals shown across graphs, i.e. by placing two dots in the model in Figure 1b) and Extended Data Fig. 1b-c) as well.

2) Figure 2 lacks important labels in c) and f), namely a clear indication of which row of images is DNA-PAINT and which is RESI and labeled scale bars. The nature of the rows in c) and f) is clear in the Supplementary Figure 2, where PAINT is placed on top of RESI in b) and e), but less so in the main figure, especially since the RESI dots are hard to make out at first

glance. Consider adding technique labels or a description in the caption to clarify. The scale bars in c) and f) should be described in the figure caption, as done in Supplementary Figure 2.

3) The last sentence on page 8 before Figure 4 has a typographical error: (Fig. 4f, Supplementary Figure 3 **dsand** Supplementary Table 1)

4) On Page 9, NND analysis of OFA is described as including "all first to tenth NNDs, like RTX", but only the 1st – 6th NND were discussed and shown before.

5) The captions of c) and d) in Extended Figure 5 are switched with each other.

6) The manuscript is well written and structured but consider reducing the use of references to a figure, an extended figure, and a supplemental figure all at once. Some figure frames appear identically in either figure or extended figure and the supplemental material and may thus be consolidated to improve readability.

Reviewer #2

(Remarks to the Author)

The manuscript by Pachmayr et al. describes the application of a multi-target 3D RESI super-resolution microscopy technique for analyzing CD20-anti CD20 complexes on target cell surfaces. The novelty of findings in the context of antigen-antibody complex structure and its relation to type I/type II characteristics is limited (see: Science 2014;343(6176):1260-3 and Science 2020;369(6505):793-799) but the applicability of RESI technique for this kind of investigation is a value of the manuscript.

Minor comments:

Regarding the sentence in the discussion: "The inverted format only produces a partial loss of direct cytotoxicity while also showing a partial increase in CD20 oligomer formation, suggesting a continuum in both therapeutic function and CD20 oligomerization between Type I and Type II mAbs, rather than two distinct categories". - I agree that there must be a smooth transition from type II to type I therapeutic function of anti-CD20 antibodies when we consider complement activation as a type I hallmark. It is possible to enforce a strong complement activation initiated by obinutuzumab in the presence of hyperactive C2 variants (see: Kuzniewska et al. Int J Mol Sci. 2024;25:10526), which would mean that even for the acknowledged type II mAbs C1q binding followed by downstream cascade activation takes place, not precluded by the lack of hexamers. Possibly, complement inhibitors deactivate such initiation at the stage of convertases before they reach high processivity.

Regarding the sentence "Training machine learning models on RESI data could predict mAb functions based on their oligomeric patterns, offering the potential for biosimilar and generic drug screening, as well as quality control", please keep in mind the existence of so-called type III anti-CD20 antibodies, that unify type I and type II characteristics (Li et al, Blood 2009;114:5007–5015, Bornstein et al, Invest New Drugs 2010;28:561–574, Nishida et al, Int J Oncol.2011;38:335–344), which may add the complexity to the prediction of mAb function.

Reviewer #3

(Remarks to the Author)

Initial comments:

This study investigates the effect of different antibodies on CD20 clustering at intact cell membranes and at the same time demonstrates value in the authors' super-resolution fluorescence technique. The data and findings would be very challenging to obtain in perhaps any other way, at the moment, although when the authors are stressing the unique capability of RESI, they should also discuss MINFLUX (and other ...FLUX-type systems). It would also be the most convincing to show something like equivalent NN plots and analysis to Fig 1c,h from the DNA-PAINT data without RESI. The findings give new information about differences in antibody binding and CD20 clustering related to signalling in cancer treatment, and suggest a relationship with a subsequent component in one of the related cytotoxic signalling pathways.

I believe this work will be significant in the establishment of new super-resolution fluorescence microscopy methods for studying small protein complexes. It may also contribute significantly to the understanding of antibody treatments for cancer.

The manuscript could be more rigorous in places (see below), but I believe the general conclusions will remain after these points are addressed.

When giving distances indicated by the NN plots, the authors could give more defined central estimates and uncertainties. Fitting may be attempted with asymmetric functions for the NN distributions (e.g. as in Curd et al., Nano Letters, 2021).

Specific comments to address:

Can the authors show how similar the behaviour of mEGFP-CD20, introduced into the cells, and CD20 are?

p2: “traditional” super-resolution microscopy methods

Which ones are these, specifically, that would not be useful for this study?

p2: “single-protein resolution”

This is not impressive by itself, only when the single proteins are close together – in the context of the CD20 dimers, particularly, here. Also the resolution of the two ALFA-tagged sites on the mAbs seems impressive (but is within-protein)! It may be good to clarify this language here (it is already appropriately specific to the receptor-mAb complex in the abstract).
- The first reference to RESI would also be useful here.

Fig. 1:

- It would be helpful to add “Type I” and “Type II” to assist the reader in Fig. 1b.
- The detail in panel c is very small, so that the various symbols given are not clearly distinguishable. This figure may need enlarging.

p3:

A little more explanation of RESI may be good, but replacing “separating” with “labeling” may be sufficient.

p3: “Angstrom resolution”

While this is the headline from ref. 15, this isn’t the case in all experiments, and it would be worth clarifying the situation in this experiment for the reader in the text at some point. ~0.6 nm s.d. precision is given in Fig. 1, which would give 1.4 nm FWHM if visualised with an appropriate spread for the uncertainty and would preclude at least resolution of targets separated by ~1.4 nm.

p4: RESI precision – it would be good to mention that this precision is obtained when orthogonal labeling strands are on nearby proteins, so in a specific proportion of cases, but not every time.

p4: “molecular resolution image”

I would prefer “an image resolving individual molecules”, which would be more specific to this case, rather than a resolution claim that may sound more general, but I will leave this to the editors to decide on.

Fig. 2:

- Much of this is low contrast and hard to see.
- What is the scale bar for c, f?
- Depending on the scale of f, this is a nice demonstration of RESI.

Sup. Fig. 1:

- Same contrast and visibility issue as Fig. 2

p5: The axial distances may be more clearly determined and given, as suggested, rather than ~30 nm and “similar” distance.

Fig. 3:

- I appreciate the high-precision nature of the RESI localisations and the benefit of seeing a larger ROI, but the specks of colour are small again. I just highlight this for the editors.
- I assume the grey circles and to do with the use of DBSCAN, but an explanation of these is needed.
- Understanding the “normalization” in the Methods is essential here. Can the reader be taken through the hidden steps to adjust for labelling efficiency and antibodies per ALFA-Nb signal, to be convincing about this data? Seeing 0.5 OBZ per cluster is confusing to start with. Similarly, the red slope in panel b is not 1:2.
- 3g, i: The ratios in g do not appear to match the ratios in i. Why is this?

A new supplementary figure (or two) showing the simulations for Fig. 3, Ext. Data Fig. 2 and Fig. 4 would be very helpful.

Suppl. Fig. 2: What data is this figure from? It seems very close to that of Fig. 3 (except for Suppl. Fig. 2c – is the y-axis mislabelled?). It is some kind of repeat?

p6: Molecular ratios – A note somewhere on how the presence of some pairs of proteins with the same RESI labeling strand affects this or does not affect it would be helpful.

p6: Distances – I assume these are from projections onto the XY plane, and Z information has been disregarded here, both in measurements and simulations. That seems reasonable, but it would be good to specify.

p6: Comparison with CSR distribution: What intensities were used in the CSR simulations (Ext. Data Fig. 2), and how were they chosen?

p6: Chain segment length ~23 nm – Can this be given with an uncertainty again? A fit of the 2nd NND distribution may be another option?

p6: C1q platforms

More tentative language seems appropriate, for presenting a new hypothesis when binding of C1q is not included in the study, and the specific shape of the actual chains is not extracted or described.

e.g.

- “the flexible nature of RTX-CD20 chains may explain”

- “For instance, U-shaped chains... could position ≥ 6 ... in close proximity”

- The Methods specify that the same ALFA-Nb may (usually?) be used in more than one overlapping C1q platform in Sup. Fig. 2b, to represent increased avidity. This needs stressing in Fig. 2, Sup. Fig. 2 and Ext. Data Fig. 2(h) as well.

- Can the authors say anything about whether C1q requires the antibodies to be in a strictly hexagonal arrangement or not (comparing with ref. 20?), and compare with more detail of the ALFA-Nb localisation pattern?

p7:

- CSR simulations – how was the parameter for this chosen in each case?

- NND distances – what were they, quantitatively? These peaks are not very clear in the histograms, more in the simulation results.

- Few compatible C1q binding platforms were detected for OBZ, not none.

Membrane bending:

- Can CD20 be tilted with respect to the membrane, rather than the membrane being bent?

p8: TCEs and flexibility

Why should the i-TCE (anti-CD3 Fab between OBZ Fabs) be expected to be more flexible than the c-TCE (anti-CD3 on the end of OBZ Fab), as is given as reasoning?

Fig. 4, Ext. Data Fig. 4, Suppl. Table 2: How the data in the table results in the plots is not clear.

p9: OFA distances

- Distances are given as exact (17 nm, 23 nm); estimates and uncertainties are needed.

Ext. Data Fig. 5: Panels c and d are described the other way around in the caption.

- About the C1q platform hypothesis:

“...while also forming possible C1q platforms...” or similar.

p11:

- TCEs: Results in the presence of T-cells are mentioned, but the relationship to this experiment in different cells needs explaining.

- “The inverted format”, last sentence, para 1. Is this the classical format – the inverted format of the inverted format? It reads as if it is different from i-TCE in the previous sentences.

-- If so, the “partial increase in CD20 oligomer formation” was assessed statistically but not found to be statistically significant, so care should be used in stating this.

- para. 2, sentence 1 on pathways – Personally, I would have found something like this helpful as extra background in the introduction.

- “Only with RESI...” – The only comparison is with DNA-PAINT, which should be made clear. However, it has also not been shown that these findings would not have been possible from the DNA-PAINT data (some kind of demonstration of this is suggested in initial comments, e.g. equivalent to Fig 1c,h from DNA-PAINT data).

-- Other high-precision techniques (e.g. MINFLUX, ...FLUX) may also be discussed with advantages and drawbacks.

- “revealing that Type I mAbs can form C1q platforms in chain-like arrangements...” – this has only been assessed without C1q in place and without explaining any restrictions on the geometry of the arrangement required for C1q - I think this should be more tentative, as above.

- “the shorter chain length we measured” – I think “measured” is a bit strong after all of the simulations and picking the point

with least disagreement; something like “deduced” might be ok.

p12: >10 cells per day... a powerful tool for screening

- This sounds like a low number for a powerful screening tool. Can the authors explain why this makes it a powerful screening tool, or describe it differently?

p17: CD20 chain-like simulations

- What was the “maximum number of chain segments” and how was it chosen?

Ext. Data Fig. 4 caption:

- “approaching values for” does not seem like the right phrase in either case when the i-TCE results are closer to OBZ and c-TCE than to RTX

Ext. Data Fig. 5g:

- The histogram bins should be centered on integer values, so the reader can understand the data.

Version 2:

Reviewer comments:

Reviewer #1

(Remarks to the Author)

All of my comments have been adequately addressed. In my view, the manuscript is ready for publication, pending correction of a small number of typographical errors:

The newly added sentence „Importantly, Fc-Fc interaction blockade does not change the structural RTX-CD20186 arrangements observed with RESI, suggesting that this RTX-mediated CD20-clustering and the formation of C1q-187 binding platforms is independent of RTX Fc-Fc interactions (Extended Data Fig. 4)” is missing a full stop at the end.

The caption of Extended Data Fig. 4c should read „SDS-PAGE of negative control for unspecific binding of BSA and IgG1 to Ni-NTA beads. ...“

There should be a blank before the caption of Extended Data Fig. 4f, at the moment it reads „... flexible-chain-like arrangement.f, Quantitative analysis ...“

Reviewer #2

(Remarks to the Author)

I appreciate that the Authors have added the additional sentence regarding my comment. I have no further issues to mention and recommend publishing the study.

Reviewer #3

(Remarks to the Author)

The authors have made many helpful improvements to the manuscript.

A couple of further minor changes would be beneficial before publication:

The new discussion related to STORM, DNA-PAINT and MINFLUX:

I agree with this discussion, however, it seems longer than is needed.

- It could perhaps start from "MINFLUX and related techniques achieve localization precisions of ..."

- "~100² μm² fields-of-view" - this seems relatively unusual (although true) notation for field of view size (e.g. ~100 x 100 μm), and this sentence including the parenthesis at the end is confusing (an acquisition rate would be expected following throughput, and the parenthesis may be redundant).

- The sentence afterwards ("Thus, RESI features a dynamic range...") would be better reconnected with the sentence before the new discussion.

- I recommend considering the flow of the whole Discussion section in refining this.

Uncertainties, Extended Data Fig. 3e and Extended Data Fig. 7c:

This new analysis is appropriate, but the definition used for "[statistically] significant difference" in the Methods and the definition of the asterisks in the plot should be included clearly.

Reviewer #1 (Remarks to the Author)

The manuscript of Pachmayr et al. extends the scope of resolution enhancement by sequential imaging (RESI) to multiple targets and the third dimension and uses this technique to characterize and differentiate Type I and Type II mAbs targeting CD20. RESI's capability to localize individual CD20 molecules in cell systems with sub-nanometer resolution is impressively shown and forms the basis of subsequent discussions of CD20 clustering induced by various mAbs, the colocalization of these molecules, and structural characteristics of the resulting complexes. The inclusion of modified antibodies and functional data, leading to a mechanistic insight of the relationship between CD20 oligomerization and Type I vs Type II mAbs makes this a consistent and well-rounded manuscript.

We thank the reviewer for the appreciation of our manuscript.

A 2023 hallmark study by the same group introduced RESI and already used one of the CD20 type I mAbs employed in this manuscript, Rituximab (RTX). Notably, the previous study used unmodified RTX while the present manuscript added nanobodies and RESI sequences to the RTX Fc. This notwithstanding, both studies essentially produce the same NND distribution of CD20 after RTX treatment. This point should be highlighted in the manuscript, as it shows that the employed labeling strategy does not change RTX-induced CD20 clustering, as might be expected by the substantial RESI labels used.

We are grateful to the reviewer for raising this important point. We have now emphasized this in the results section: "Notably, CD20 NND's after treatment with ALFA-tagged and untagged RTX are comparable, showing that the ALFA-tag does not affect the mAb's CD20 binding properties"

The authors demonstrate that Type I mAbs form structures termed "C1q binding platforms", while Type II mAbs can do so only to a severely reduced extent, attributing this platform formation to large U-shaped clusters. While the presented quantitative cluster analysis and NND data supports this claim, this reviewer strongly suggests performing additional experiments to gauge the role of Fc-Fc interactions in the formation of these clusters. IgG oligomerization is the accepted mechanism of specific C1q recruitment and C1 activation leading to CDC, and IgG hexamerization would be consistent with the presented number of RTX per cluster (Fig 3b), and thus should be adequately considered and discussed. This may be achieved by using CDC-enhanced RTX and OBZ point mutants that have been shown to increase IgG oligomerization via Fc-Fc interactions (e.g. E345R, E345K, or E430G) and gauging the effects on observed cluster sizes and NNDs, as was already done for c-TCE and i-TCE. IgG oligomerization inhibiting mutations may also be used to rule out significant contributions of this process to the CD20 clustering observed by RTX. Alternatively, Fc blocking agents such as a Fc binding peptide like DCAWHLGELVWCT used by Diebolder et al. (Science. 2014 March 14; 343(6176): 1260–1263) or derivatives of Staphylococcal protein A (PNAS. 2021 February 9; 118 (7) e2016772118) might be employed to reversibly block IgG oligomerization in this case. Showing actual C1q binding data to correlate with C1q platform abundances would further improve this point.

Finally, were the direct cell killing assays performed with the same labeled mAbs as the rest of the study? This point was not clear to this reviewer but may be critical to interpreting these results.

We very much appreciate the constructive feedback and thank the reviewer for a very thoughtful and detailed suggestion for a potential experimental plan.

We decided to gauge the role of Fc-Fc interactions in the formation of these clusters by blocking the Fc-Fc interactions during RTX and OBZ treatment and subsequently measuring RESI and performing cluster analysis. We chose to block the interactions using the Staphylococcal protein A derivative B (SpA-B) (PNAS. 2021 February 9; 118 (7) e2016772118), as suggested by the reviewer, that has been previously used to successfully decrease the proportion of hexamers in hexamerizing IgGs. We demonstrated the binding of SpA-B to IgG1 in pulldown assays and pre-incubated a 20-fold excess of SpA-B with RTX or OBZ before treatment. The results are now displayed in Extended Data Figure 4. Essentially, we found that there is no apparent effect of Fc-Fc interaction blockade on the CD20 as well as mAb arrangement on cells.

Furthermore, we now mention that we assessed the existence of hexameric platforms in our data: “We have previously shown that isolated hexameric circular platforms of CD20 and RTX, as postulated by Cryo-EM studies, are not compatible with our RESI data showing highly concatenated linear RTX-CD20 clusters [...]. Although circular arrangements may be present in limited amounts on the cell membrane, this suggests that an alternative structural organization leads to C1q binding platforms.”

Regarding the functional studies of the direct CD20 binding effects, these were executed with untagged versions of all antibodies. However, due to the placement of the ALFA-tag at the HC-C terminus, far away from the CD20 binding site and due to the fact that CD20 re-arrangement does not differ between untagged and tagged versions of RTX and OBZ, we believe that the ALFA-tag does not affect CD20 binding capabilities. We have further increased clarity by adding “untagged versions of mAbs” in the figure description.

Specific comments:

1) Figure 1 shows models of antibody-mediated CD20 arrangements (b) and actual RESI localizations (c) side by side. The model uses one colored dot to represent a monomer, while in the real data two localizations are recorded per antibody (but not CD20) as per the labeling strategy. Consider standardizing the number of signals shown across graphs, i.e. by placing two dots in the model in Figure 1b) and Extended Data Fig. 1b-c) as well.

We decided to keep Fig. 1b with one “dot” representing one antibody as we only introduce the labeling strategy in Fig. 1c. In order to further clarify this point, we now added this sentence to the figure caption: “CD20 is labeled in 1:1 stoichiometry and mAbs are labeled in a 2:1 stoichiometry.”

2) Figure 2 lacks important labels in c) and f), namely a clear indication of which row of images is DNA-PAINT and which is RESI and labeled scale bars. The nature of the rows in c) and f) is clear in the Supplementary Figure 2, where PAINT is placed on top of RESI in b) and e), but less so in the main figure, especially since the RESI dots are hard to make out at first glance. Consider adding technique

labels or a description in the caption to clarify. The scale bars in c) and f) should be described in the figure caption, as done in Supplementary Figure 2.

Thank you for pointing out that DNA-PAINT and RESI images are hard to distinguish in Figure 2. We now put one label per row of images in Fig. 2c and f and added the length of the scale bars to the figure caption.

3) The last sentence on page 8 before Figure 4 has a typographical error: (Fig. 4f, Supplementary Figure 3 dsand Supplementary Table 1)

Thank you for spotting the error, we changed the text accordingly.

4) On Page 9, NND analysis of OFA is described as including “all first to tenth NNDs, like RTX”, but only the 1st – 6th NND were discussed and shown before.

You are right, we only display the first to sixth NNDs here, we changed the text accordingly.

5) The captions of c) and d) in Extended Figure 5 are switched with each other.

Thank you for noticing that mistake. We changed the figure caption accordingly.

6) The manuscript is well written and structured but consider reducing the use of references to a figure, an extended figure, and a supplemental figure all at once. Some figure frames appear identically in either figure or extended figure and the supplemental material and may thus be consolidated to improve readability.

Thank you for your thoughtful feedback. We understand the concern regarding the redundancy of referencing figures, extended figures, and supplemental figures. However, we believe that the current structure of the manuscript, where the main figure presents one replicate, and the supplemental materials include additional replicates, provides clarity and allows for a more comprehensive understanding of the data. The main figure highlights key findings, while the supplemental figures and tables offer supporting information, including the full set of data, for readers who wish to explore further.

We believe that consolidating these figures would compromise the ability to clearly convey the variability and robustness of our results. Therefore, we have decided to maintain the current organization. We hope this explanation clarifies the rationale behind our figure structure.

Reviewer #2 (Remarks to the Author)

The manuscript by Pachmayr et al. describes the application of a multi-target 3D RESI super-resolution microscopy technique for analyzing CD20-anti CD20 complexes on target cell surfaces. The novelty of findings in the context of antigen-antibody complex structure and its relation to type I/type II characteristics is limited (see: Science 2014;343(6176):1260-3 and Science 2020;369(6505):793-799) but the applicability of RESI technique for this kind of investigation is a value of the manuscript.

We thank the reviewer for the appreciation of our manuscript.

Minor comments:

Regarding the sentence in the discussion: "The inverted format only produces a partial loss of direct cytotoxicity while also showing a partial increase in CD20 oligomer formation, suggesting a continuum in both therapeutic function and CD20 oligomerization between Type I and Type II mAbs, rather than two distinct categories". - I agree that there must be a smooth transition from type II to type I therapeutic function of anti-CD20 antibodies when we consider complement activation as a type I hallmark. It is possible to enforce a strong complement activation initiated by obinutuzumab in the presence of hyperactive C2 variants (see: Kuzniewska et al. Int J Mol Sci. 2024;25:10526), which would mean that even for the acknowledged type II mAbs C1q binding followed by downstream cascade activation takes place, not precluded by the lack of hexamers. Possibly, complement inhibitors deactivate such initiation at the stage of convertases before they reach high processivity.

Regarding the sentence "Training machine learning models on RESI data could predict mAb functions based on their oligomeric patterns, offering the potential for biosimilar and generic drug screening, as well as quality control", please keep in mind the existence of so-called type III anti-CD20 antibodies, that unify type I and type II characteristics (Li et al, Blood 2009;114,5007–5015, Bornstein et al, Invest New Drugs 2010;28:561–574, Nishida et al, Int J Oncol.2011;38:335–344), which may add the complexity to the prediction of mAb function.

Thank you for raising the point about the complexity introduced by anti-CD20 antibodies unifying the characteristics of type I and type II antibodies. We agree that the presence of such antibodies adds an additional layer of complexity to predicting mAb functions based solely on their oligomeric patterns. To address this, we have included a sentence in the manuscript stating: "To further refine the relationship between structure and function of anti-CD20 mAbs, future studies could investigate those mAbs that have been shown to unify Type I and Type II functionalities [Li et al, Blood 2009;114,5007–5015]."

Reviewer #3 (Remarks to the Author)

Initial comments:

This study investigates the effect of different antibodies on CD20 clustering at intact cell membranes and at the same time demonstrates value in the authors' super-resolution fluorescence technique. The data and findings would be very challenging to obtain in perhaps any other way, at the moment, although when the authors are stressing the unique capability of RESI, they should also discuss MINFLUX (and other ...FLUX-type systems).

Thank you for raising this point. We now mention MINFLUX and other FLUX-type systems in the Discussion: "Although MINFLUX and related techniques achieve sub-5 nm resolution, their current throughput remains limited for whole-cell imaging."

It would also be the most convincing to show something like equivalent NN plots and analysis to Fig 1c,h from the DNA-PAINT data without RESI.

We appreciate this suggestion and have now added Extended Data Figure 2 to demonstrate the differences of NNDs detected with DNA-PAINT and RESI.

The findings give new information about differences in antibody binding and CD20 clustering related to signalling in cancer treatment, and suggest a relationship with a subsequent component in one of the related cytotoxic signalling pathways.

I believe this work will be significant in the establishment of new super-resolution fluorescence microscopy methods for studying small protein complexes. It may also contribute significantly to the understanding of antibody treatments for cancer.

We thank the reviewer for the appreciation of our manuscript.

The manuscript could be more rigorous in places (see below), but I believe the general conclusions will remain after these points are addressed.

When giving distances indicated by the NN plots, the authors could give more defined central estimates and uncertainties. Fitting may be attempted with asymmetric functions for the NN distributions (e.g. as in Curd et al., Nano Letters, 2021).

Specific comments to address:

Can the authors show how similar the behaviour of mEGFP-CD20, introduced into the cells, and

CD20 are?

We thank the reviewer for raising this important point. Unfortunately, we were not able to directly label CD20 for RESI, as there is no high affinity and high specificity monovalent binder available. We did however introduce the much smaller ALFA-tag instead of the mEGFP-tag to CD20 and imaged the CD20 resting state. When comparing the monomer and dimer ratios of ALFA-tagged with mEGFP-tagged CD20, we did not detect any differences in the resting state, indicating that the size of the GFP-tag does not influence CD20 behavior in our experiments.

p2: “traditional” super-resolution microscopy methods

Which ones are these, specifically, that would not be useful for this study?

*We clarified this now in this revised sentence: “Cryo-EM, mass spectrometry and traditional super-resolution microscopy methods, such as stimulated emission depletion, (direct) stochastic optical reconstruction or photoactivated localization microscopy, are limited because they cannot achieve single-protein resolution *in situ* for dense assemblies [...].”*

p2: “single-protein resolution”

This is not impressive by itself, only when the single proteins are close together – in the context of the CD20 dimers, particularly, here. Also the resolution of the two ALFA-tagged sites on the mAbs seems impressive (but is within-protein)! It may be good to clarify this language here (it is already appropriately specific to the receptor-mAb complex in the abstract).

- The first reference to RESI would also be useful here.

We thank the reviewer for pointing out that we actually resolve intramolecular distances and specified the language for Figure 2f accordingly.: “Excitingly, we were able to resolve two ALFA-Nbs bound to a single OBZ mAb, thus visualizing intramolecular distances within a protein (Fig. 3f, top).” We also added a first reference to RESI.

Fig. 1:

- It would be helpful to add “Type I” and “Type II” to assist the reader in Fig. 1b.

- The detail in panel c is very small, so that the various symbols given are not clearly distinguishable. This figure may need enlarging.

We agree with the reviewer and added Type I and Type II to Fig. 1b.

p3:

A little more explanation of RESI may be good, but replacing “separating” with “labeling” may be sufficient.

We changed the sentence accordingly: “After mAb treatment and fixation, we performed stochastic separation by labeling with four orthogonal DNA sequences per target according to Fig. 1c.”

p3: “Angstrom resolution”

While this is the headline from ref. 15, this isn’t the case in all experiments, and it would be worth clarifying the situation in this experiment for the reader in the text at some point. ~0.6 nm s.d. precision is given in Fig. 1, which would give 1.4 nm FWHM if visualised with an appropriate spread for the uncertainty and would preclude at least resolution of targets separated by ~1.4 nm.

This is true, we reach approx. 1.4 nm FWHM in 3D measurements. We do not claim that we achieve Ångström resolution in our measurements. We believe that giving exact numbers for resolution based on the achieved precision in these specific measurements is not possible, since we are still limited by the label size. ALFA-tag and ALFA-nanobody are ~ 2nm in size and the GFP-tag and cognate nanobody are ~5 nm in size, limiting the achievable resolution. We now write: We achieved an effective resolution of ~2 to ~5 nm, limited by the size of the ALFA-tag-Nb or the mEGFP-tag-Nb complex, respectively.”

p4: RESI precision – it would be good to mention that this precision is obtained when orthogonal labeling strands are on nearby proteins, so in a specific proportion of cases, but not every time.

We appreciate the feedback from the reviewer. We have now added a sentence to clarify this: “In order to achieve this, two adjacent molecules have to be labeled with orthogonal docking strands.”

p4: “molecular resolution image”

I would prefer “an image resolving individual molecules”, which would be more specific to this case, rather than a resolution claim that may sound more general, but I will leave this to the editors to decide on.

We appreciate the reviewer's perspective and have changed the sentence accordingly.

Fig. 2:

- Much of this is low contrast and hard to see.
- What is the scale bar for c, f?
- Depending on the scale of f, this is a nice demonstration of RESI.

Thank you for pointing out this issue, we now added the length of the scale bars to the figure captions of Fig.2c and f.

Sup. Fig. 1:

- Same contrast and visibility issue as Fig. 2

p5: The axial distances may be more clearly determined and given, as suggested, rather than ~30 nm and “similar” distance.

Thank you for pointing this out. We now determine mean and std for the distances and have added this to the main text and figure caption.

Fig. 3:

- I appreciate the high-precision nature of the RESI localisations and the benefit of seeing a larger ROI, but the specks of colour are small again. I just highlight this for the editors.

- I assume the grey circles and to do with the use of DBSCAN, but an explanation of these is needed. *We now added “DBSCAN (clusters marked in gray)” to the figure caption to address this.*

- Understanding the “normalization” in the Methods is essential here. Can the reader be taken through the hidden steps to adjust for labelling efficiency and antibodies per ALFA-Nb signal, to be convincing about this data? Seeing 0.5 OBZ per cluster is confusing to start with. Similarly, the red slope in panel b is not 1:2.

In the ratio analysis, we decided to treat 2 ALFA-Nbs within a cluster as one RTX or OBZ. Especially in the case of RTX, this nicely shows the linear correlation of RTX and CD20. Due to the labeling efficiency of 40% of the ALFA-Nb, this can result in 0.5 OBZ per cluster. We now clearly state this in the figure caption: “Two ALFA-Nbs correspond to one RTX/OBZ per cluster.”

We agree that understanding the normalization is essential for the data. We already addressed this in the Methods “Ratio analysis” and now referred directly to this in the figure caption and main text.

- 3g, i: The ratios in g do not appear to match the ratios in i. Why is this?

The ratios of 3g are directly determined from the analyzed data and are not labeling efficiency corrected. The proportions of complexes in 3i are determined from simulations taking into account the labeling efficiency. We now state this clearly in the figure caption: “Simulations with CD20 monomers, dimers, trimers and tetramers, taking into account the labeling efficiency of 50 % of the GFP-Nb, for this representative cell result in 35 % monomers, 48 % dimers, 10 % trimers and 7 % tetramers after OBZ treatment.”

A new supplementary figure (or two) showing the simulations for Fig. 3, Ext. Data Fig. 2 and Fig. 4 would be very helpful.

We now added one supplementary figure displaying the data of the chain simulation for RTX treatment and the oligomer simulation for OBZ treatment (Fig. 3, Ext. Data Fig. 2) and one supplementary figure displaying the oligomer simulation for i-TCE and c-TCE treatment of Fig.4.

Suppl. Fig. 2: What data is this figure from? It seems very close to that of Fig. 3 (except for Suppl. Fig. 2c – is the y-axis mis-labelled?). It is some kind of repeat?

The data of the Suppl. Fig. is a replicate, we now indicate that in the figure caption. We changed the y-axis label in Suppl. Fig. 2c to the correct values.

p6: Molecular ratios – A note somewhere on how the presence of some pairs of proteins with the same RESI labeling strand affects this or does not affect it would be helpful.

The presence of some pairs with the same docking strand can lead to an underestimation of higher-order oligomers, if all of them are DNA-PAINT unresolvable. However, due to a high flexibility of the labeling probe, not all molecules within an oligomer are DNA-PAINT unresolvable so that this effect is mitigated. We assessed this in the RESI paper [<https://doi.org/10.1038/s41586-023-05925-9>] and now mention this in the main text. “Even though the detection efficiency of oligomers in RESI may be reduced due to two adjacent DNA-PAINT-unresolvable molecules labeled with the same DNA docking strand sequence, this cannot explain a trimer and tetramer proportion below 30 % [<https://doi.org/10.1038/s41586-023-05925-9>].”

p6: Distances – I assume these are from projections onto the XY plane, and Z information has been disregarded here, both in measurements and simulations. That seems reasonable, but it would be good to specify.

That is correct, the nearest-neighbor distances are quantified in 2D. We have now specified this in the figure caption.

p6: Comparison with CSR distribution: What intensities were used in the CSR simulations (Ext. Data Fig. 2), and how were they chosen?

The CSR distributions are simulated at the same molecular densities that were measured in the cell images, as specified in the methods “Simulation of monomers”.

p6: Chain segment length ~23 nm – Can this be given with an uncertainty again? A fit of the 2nd NND distribution may be another option?

We now performed Monte-Carlo simulations to assess the uncertainty of the chain segment lengths and show this in the same Extended data figures as before.

p6: C1q platforms

More tentative language seems appropriate, for presenting a new hypothesis when binding of C1q is not included in the study, and the specific shape of the actual chains is not extracted or described.

e.g.

- “the flexible nature of RTX-CD20 chains _may_ explain”

- “_For instance_, U-shaped chains... could position ≥ 6 ... in close proximity”

We now changed the text to the following:

“Furthermore, the flexible nature of RTX-CD20 chains can explain how RTX assemblies organize to form C1q binding platforms.”

“_For instance_, U-shaped chains of 6 CD20 dimers could position 6 RTX-Fc domains in close proximity to allow for efficient C1q binding.”

- The Methods specify that the same ALFA-Nb may (usually?) be used in more than one overlapping C1q platform in Sup. Fig. 2b, to represent increased avidity. This needs stressing in Fig. 2, Sup. Fig. 2 and Ext. Data Fig. 2(h) as well.

We now added a sentence in the Fig.3 that clearly states that all binding possibilities are counted as C1q platform. “The number of hexameric RTX platforms was determined by counting each possible binding configuration of C1q, in order to represent the apparent gain in avidity, thus individual RTX molecules can be part of several distinct platforms.”

- Can the authors say anything about whether C1q requires the antibodies to be in a strictly hexagonal arrangement or not (comparing with ref. 20?), and compare with more detail of the ALFA-Nb localisation pattern?

Unfortunately, we cannot directly deduct hexagonal patterns from the data. The flexibility of both IgGs and ALFA-labels prevents that. C1q flexibility as well as IgG flexibility suggest that the hexagonal arrangement is not the most important feature but rather the proximity to increase avidity plays a role.

p7:

- CSR simulations – how was the parameter for this chosen in each case?

CSR simulations were performed as explained in the Methods “CD20 low order oligomerization simulations” by simulating only monomers that are CSR distributed.

- NND distances – what were they, quantitatively? These peaks are not very clear in the histograms, more in the simulation results.

They were below 20 nm. We now mention this in the main text: “This comparison revealed specific peaks at distances below 20 nm for 1st to 3rd NNDs, which cannot be explained by a pure CSR distribution. “

- Few compatible C1q binding platforms were detected for OBZ, not none.

We changed it to: “Accordingly, we did detect only a few compatible C1q binding platforms for OBZ (Extended Data Fig. 3h) as compared to RTX. ”

Membrane bending:

- Can CD20 be tilted with respect to the membrane, rather than the membrane being bent?

Yes, that can happen, we changed the text accordingly to: “...suggesting local nanoscale bending of the cell membrane, although we cannot exclude that tilting of the whole complex contributes to this phenomenon.”

p8: TCEs and flexibility

Why should the i-TCE (anti-CD3 Fab between OBZ Fabs) be expected to be more flexible than the c-TCE (anti-CD3 on the end of OBZ Fab), as is given as reasoning?

The flexible linkers between anti-CD3-Fab and anti-CD20-Fab as well as the presence of anti-CD3-Fab between the two CD20-Fabs in inverted TCE (i-TCE) allow for a higher degree of freedom of CD20 binding when compared with the classical TCE (c-TCE).

Fig. 4, Ext. Data Fig. 4, Suppl. Table 2: How the data in the table results in the plots is not clear.

We now refer to the Supplementary Table 2 in the Figure captions of Fig. 4 and Extended Data Fig.4 (now 6). These correspond to the different sheets in the excel table, which we now explicitly write out in the caption.

p9: OFA distances

- Distances are given as exact (17 nm, 23 nm); estimates and uncertainties are needed.

We performed Monte Carlo simulations to estimate the uncertainties of the chain-model for different distances. The results are now plotted in Extended Data Fig. 3 for RTX and Extended Data Fig. 7 for OFA.

Ext. Data Fig. 5: Panels c and d are described the other way around in the caption.

We apologize for this oversight and changed the text accordingly.

- About the C1q platform hypothesis:
"...while also forming _possible_ C1q platforms..." or similar.

We changed the text to "...while also forming platforms compatible with C1q binding."

p11:

- TCEs: Results in the presence of T-cells are mentioned, but the relationship to this experiment in different cells needs explaining.

*We changed the text: "Interestingly, when only investigating the CD20-mediated direct cytotoxicity, **independent of T cell mediated effects**, we found that **i-TCE** reduces the original direct cytotoxicity of **c-TCE** while it increases the formation of CD20 trimers and tetramers."*

- "The inverted format", last sentence, para 1. Is this the classical format – the inverted format of the inverted format? It reads as if it is different from i-TCE in the previous sentences.

*Sorry for the lack of clarity in the text. We changed the text to: "Interestingly, when only investigating the CD20-mediated direct cytotoxicity, **independent of T cell mediated effects**, we found that **i-TCE** reduces the original direct cytotoxicity of **c-TCE** while it increases the formation of CD20 trimers and tetramers."*

-- If so, the "partial increase in CD20 oligomer formation" was assessed statistically but not found to be statistically significant, so care should be used in stating this.

This should be solved now based on the previous clarification.

- para. 2, sentence 1 on pathways – Personally, I would have found something like this helpful as extra background in the introduction.

We think it is appropriate to mention this only in the discussion to keep it as concise as possible.

- “Only with RESI...” – The only comparison is with DNA-PAINT, which should be made clear. However, it has also not been shown that these findings would not have been possible from the DNA-PAINT data (some kind of demonstration of this is suggested in initial comments, e.g. equivalent to Fig 1c,h from DNA-PAINT data).

We added Extended Data Fig. 2 to assess the advantage of RESI over DNA-PAINT.

- Other high-precision techniques (e.g. MINFLUX, ...FLUX) may also be discussed with advantages and drawbacks.

We added a sentence in the discussion to compare DNA-PAINT and RESI with MINFLUX (see above).

- “revealing that Type I mAbs can form C1q platforms in chain-like arrangements...” – this has only been assessed without C1q in place and without explaining any restrictions on the geometry of the arrangement required for C1q - I think this should be more tentative, as above.

We changed it to: “...revealing that Type I mAbs can form platforms compatible with C1q binding within chain-like arrangements...”

- “the shorter chain length we measured” – I think “measured” is a bit strong after all of the simulations and picking the point with least disagreement; something like “deduced” might be ok.

We changed it to “deduced” accordingly.

p12: >10 cells per day... a powerful tool for screening

- This sounds like a low number for a powerful screening tool. Can the authors explain why this makes it a powerful screening tool, or describe it differently?

We do agree, however we believed that this is currently the most powerful screening tool at this resolution. Thus we changed the text accordingly: “The current throughput of RESI allows imaging >10 cells per day, making it a powerful tool for screening therapeutic mAb candidates at molecular resolution.”

p17: CD20 chain-like simulations

- What was the “maximum number of chain segments” and how was it chosen?

It was chosen according to the experimentally determined cluster size. Each chain was simulated separately with a length according to the frequency of experimentally determined cluster sizes. For

details see Methods “chain simulation”.

Ext. Data Fig. 4 caption:

- “approaching values for” does not seem like the right phrase in either case when the i-TCE results are closer to OBZ and c-TCE than to RTX

We changed it to “Trending toward values for” where appropriate.

Ext. Data Fig. 5g:

- The histogram bins should be centered on integer values, so the reader can understand the data.

We changed the Extended Data Fig. 5 (now 7) accordingly.

Reviewer #1 (Remarks to the Author)

The manuscript of Pachmayr et al. extends the scope of resolution enhancement by sequential imaging (RESI) to multiple targets and the third dimension and uses this technique to characterize and differentiate Type I and Type II mAbs targeting CD20. RESI's capability to localize individual CD20 molecules in cell systems with sub-nanometer resolution is impressively shown and forms the basis of subsequent discussions of CD20 clustering induced by various mAbs, the colocalization of these molecules, and structural characteristics of the resulting complexes. The inclusion of modified antibodies and functional data, leading to a mechanistic insight of the relationship between CD20 oligomerization and Type I vs Type II mAbs makes this a consistent and well-rounded manuscript.

We thank the reviewer for the appreciation of our manuscript.

A 2023 hallmark study by the same group introduced RESI and already used one of the CD20 type I mAbs employed in this manuscript, Rituximab (RTX). Notably, the previous study used unmodified RTX while the present manuscript added nanobodies and RESI sequences to the RTX Fc. This notwithstanding, both studies essentially produce the same NND distribution of CD20 after RTX treatment. This point should be highlighted in the manuscript, as it shows that the employed labeling strategy does not change RTX-induced CD20 clustering, as might be expected by the substantial RESI labels used.

We are grateful to the reviewer for raising this important point. We have now emphasized this in the results section: "Notably, CD20 NND's after treatment with ALFA-tagged and untagged RTX are comparable, showing that the ALFA-tag does not affect the mAb's CD20 binding properties"

The authors demonstrate that Type I mAbs form structures termed "C1q binding platforms", while Type II mAbs can do so only to a severely reduced extent, attributing this platform formation to large U-shaped clusters. While the presented quantitative cluster analysis and NND data supports this claim, this reviewer strongly suggests performing additional experiments to gauge the role of Fc-Fc interactions in the formation of these clusters. IgG oligomerization is the accepted mechanism of specific C1q recruitment and C1 activation leading to CDC, and IgG hexamerization would be consistent with the presented number of RTX per cluster (Fig 3b), and thus should be adequately considered and discussed. This may be achieved by using CDC-enhanced RTX and OBZ point mutants that have been shown to increase IgG oligomerization via Fc-Fc interactions (e.g. E345R, E345K, or E430G) and gauging the effects on observed cluster sizes and NNDs, as was already done for c-TCE and i-TCE. IgG oligomerization inhibiting mutations may also be used to rule out significant contributions of this process to the CD20 clustering observed by RTX. Alternatively, Fc blocking agents such as a Fc binding peptide like DCAWHLGELVWCT used by Diebolder et al. (Science. 2014 March 14; 343(6176): 1260–1263) or derivatives of Staphylococcal protein A (PNAS. 2021 February 9; 118 (7) e2016772118) might be employed to reversibly block IgG oligomerization in this case. Showing actual C1q binding data to correlate with C1q platform abundances would further improve this point.

Finally, were the direct cell killing assays performed with the same labeled mAbs as the rest of the study? This point was not clear to this reviewer but may be critical to interpreting these results.

We very much appreciate the constructive feedback and thank the reviewer for a very thoughtful and detailed suggestion for a potential experimental plan.

We decided to gauge the role of Fc-Fc interactions in the formation of these clusters by blocking the Fc-Fc interactions during RTX and OBZ treatment and subsequently measuring RESI and performing cluster analysis. We chose to block the interactions using the Staphylococcal protein A derivative B (SpA-B) (PNAS. 2021 February 9; 118 (7) e2016772118), as suggested by the reviewer, that has been previously used to successfully decrease the proportion of hexamers in hexamerizing IgGs. We demonstrated the binding of SpA-B to IgG1 in pulldown assays and pre-incubated a 20-fold excess of SpA-B with RTX or OBZ before treatment. The results are now displayed in Extended Data Figure 4. Essentially, we found that there is no apparent effect of Fc-Fc interaction blockade on the CD20 as well as mAb arrangement on cells.

Furthermore, we now mention that we assessed the existence of hexameric platforms in our data: “We have previously shown that isolated hexameric circular platforms of CD20 and RTX, as postulated by Cryo-EM studies, are not compatible with our RESI data showing highly concatenated linear RTX-CD20 clusters [...]. Although circular arrangements may be present in limited amounts on the cell membrane, this suggests that an alternative structural organization leads to C1q binding platforms.”

Regarding the functional studies of the direct CD20 binding effects, these were executed with untagged versions of all antibodies. However, due to the placement of the ALFA-tag at the HC-C terminus, far away from the CD20 binding site and due to the fact that CD20 re-arrangement does not differ between untagged and tagged versions of RTX and OBZ, we believe that the ALFA-tag does not affect CD20 binding capabilities. We have further increased clarity by adding “untagged versions of mAbs” in the figure description.

Specific comments:

1) Figure 1 shows models of antibody-mediated CD20 arrangements (b) and actual RESI localizations (c) side by side. The model uses one colored dot to represent a monomer, while in the real data two localizations are recorded per antibody (but not CD20) as per the labeling strategy. Consider standardizing the number of signals shown across graphs, i.e. by placing two dots in the model in Figure 1b) and Extended Data Fig. 1b-c) as well.

We decided to keep Fig. 1b with one “dot” representing one antibody as we only introduce the labeling strategy in Fig. 1c. In order to further clarify this point, we now added this sentence to the figure caption: “CD20 is labeled in 1:1 stoichiometry and mAbs are labeled in a 2:1 stoichiometry.”

2) Figure 2 lacks important labels in c) and f), namely a clear indication of which row of images is DNA-PAINT and which is RESI and labeled scale bars. The nature of the rows in c) and f) is clear in the Supplementary Figure 2, where PAINT is placed on top of RESI in b) and e), but less so in the main figure, especially since the RESI dots are hard to make out at first glance. Consider adding technique

labels or a description in the caption to clarify. The scale bars in c) and f) should be described in the figure caption, as done in Supplementary Figure 2.

Thank you for pointing out that DNA-PAINT and RESI images are hard to distinguish in Figure 2. We now put one label per row of images in Fig. 2c and f and added the length of the scale bars to the figure caption.

3) The last sentence on page 8 before Figure 4 has a typographical error: (Fig. 4f, Supplementary Figure 3 dsand Supplementary Table 1)

Thank you for spotting the error, we changed the text accordingly.

4) On Page 9, NND analysis of OFA is described as including “all first to tenth NNDs, like RTX”, but only the 1st – 6th NND were discussed and shown before.

You are right, we only display the first to sixth NNDs here, we changed the text accordingly.

5) The captions of c) and d) in Extended Figure 5 are switched with each other.

Thank you for noticing that mistake. We changed the figure caption accordingly.

6) The manuscript is well written and structured but consider reducing the use of references to a figure, an extended figure, and a supplemental figure all at once. Some figure frames appear identically in either figure or extended figure and the supplemental material and may thus be consolidated to improve readability.

Thank you for your thoughtful feedback. We understand the concern regarding the redundancy of referencing figures, extended figures, and supplemental figures. However, we believe that the current structure of the manuscript, where the main figure presents one replicate, and the supplemental materials include additional replicates, provides clarity and allows for a more comprehensive understanding of the data. The main figure highlights key findings, while the supplemental figures and tables offer supporting information, including the full set of data, for readers who wish to explore further.

We believe that consolidating these figures would compromise the ability to clearly convey the variability and robustness of our results. Therefore, we have decided to maintain the current organization. We hope this explanation clarifies the rationale behind our figure structure.

Reviewer #2 (Remarks to the Author)

The manuscript by Pachmayr et al. describes the application of a multi-target 3D RESI super-resolution microscopy technique for analyzing CD20-anti CD20 complexes on target cell surfaces. The novelty of findings in the context of antigen-antibody complex structure and its relation to type I/type II characteristics is limited (see: Science 2014;343(6176):1260-3 and Science 2020;369(6505):793-799) but the applicability of RESI technique for this kind of investigation is a value of the manuscript.

We thank the reviewer for the appreciation of our manuscript.

Minor comments:

Regarding the sentence in the discussion: "The inverted format only produces a partial loss of direct cytotoxicity while also showing a partial increase in CD20 oligomer formation, suggesting a continuum in both therapeutic function and CD20 oligomerization between Type I and Type II mAbs, rather than two distinct categories". - I agree that there must be a smooth transition from type II to type I therapeutic function of anti-CD20 antibodies when we consider complement activation as a type I hallmark. It is possible to enforce a strong complement activation initiated by obinutuzumab in the presence of hyperactive C2 variants (see: Kuzniewska et al. Int J Mol Sci. 2024;25:10526), which would mean that even for the acknowledged type II mAbs C1q binding followed by downstream cascade activation takes place, not precluded by the lack of hexamers. Possibly, complement inhibitors deactivate such initiation at the stage of convertases before they reach high processivity.

Regarding the sentence "Training machine learning models on RESI data could predict mAb functions based on their oligomeric patterns, offering the potential for biosimilar and generic drug screening, as well as quality control", please keep in mind the existence of so-called type III anti-CD20 antibodies, that unify type I and type II characteristics (Li et al, Blood 2009;114,5007–5015, Bornstein et al, Invest New Drugs 2010;28:561–574, Nishida et al, Int J Oncol.2011;38:335–344), which may add the complexity to the prediction of mAb function.

Thank you for raising the point about the complexity introduced by anti-CD20 antibodies unifying the characteristics of type I and type II antibodies. We agree that the presence of such antibodies adds an additional layer of complexity to predicting mAb functions based solely on their oligomeric patterns. To address this, we have included a sentence in the manuscript stating: "To further refine the relationship between structure and function of anti-CD20 mAbs, future studies could investigate those mAbs that have been shown to unify Type I and Type II functionalities [Li et al, Blood 2009;114,5007–5015]."

Reviewer #3 (Remarks to the Author)

Initial comments:

This study investigates the effect of different antibodies on CD20 clustering at intact cell membranes and at the same time demonstrates value in the authors' super-resolution fluorescence technique. The data and findings would be very challenging to obtain in perhaps any other way, at the moment, although when the authors are stressing the unique capability of RESI, they should also discuss MINFLUX (and other ...FLUX-type systems).

Thank you for raising this point. We now mention MINFLUX and other FLUX-type systems in the Discussion, please see the specific comment addressed below.

It would also be the most convincing to show something like equivalent NN plots and analysis to Fig 1c,h from the DNA-PAINT data without RESI.

We appreciate this suggestion and have now added Extended Data Figure 2 to demonstrate the differences of NNDs detected with DNA-PAINT and RESI.

The findings give new information about differences in antibody binding and CD20 clustering related to signalling in cancer treatment, and suggest a relationship with a subsequent component in one of the related cytotoxic signalling pathways.

I believe this work will be significant in the establishment of new super-resolution fluorescence microscopy methods for studying small protein complexes. It may also contribute significantly to the understanding of antibody treatments for cancer.

We thank the reviewer for the appreciation of our manuscript.

The manuscript could be more rigorous in places (see below), but I believe the general conclusions will remain after these points are addressed.

When giving distances indicated by the NN plots, the authors could give more defined central estimates and uncertainties. Fitting may be attempted with asymmetric functions for the NN distributions (e.g. as in Curd et al., Nano Letters, 2021).

Specific comments to address:

Can the authors show how similar the behaviour of mEGFP-CD20, introduced into the cells, and CD20 are?

We thank the reviewer for raising this important point. Unfortunately, we were not able to directly label CD20 for RESI, as there is no high affinity and high specificity monovalent binder available. We did however introduce the much smaller ALFA-tag instead of the mEGFP-tag to CD20 and imaged the CD20 resting state. When comparing the monomer and dimer ratios of ALFA-tagged with mEGFP-tagged CD20, we did not detect any differences in the resting state, indicating that the size of the GFP-tag does not influence CD20 behavior in our experiments.

p2: “traditional” super-resolution microscopy methods

Which ones are these, specifically, that would not be useful for this study?

We clarified this now in this revised sentence: “Cryo-EM, mass spectrometry and traditional super-resolution microscopy methods, such as stimulated emission depletion, (direct) stochastic optical reconstruction or photoactivated localization microscopy, are limited because they cannot achieve single-protein resolution in situ for dense assemblies [...]”

p2: “single-protein resolution”

This is not impressive by itself, only when the single proteins are close together – in the context of the CD20 dimers, particularly, here. Also the resolution of the two ALFA-tagged sites on the mAbs seems impressive (but is within-protein)! It may be good to clarify this language here (it is already appropriately specific to the receptor-mAb complex in the abstract).

- The first reference to RESI would also be useful here.

We thank the reviewer for pointing out that we actually resolve intramolecular distances and specified the language for Figure 2f accordingly.: “Excitingly, we were able to resolve two ALFA-Nbs bound to a single OBZ mAb, thus visualizing intramolecular distances within a protein (Fig. 3f, top).” We also added a first reference to RESI.

Fig. 1:

- It would be helpful to add “Type I” and “Type II” to assist the reader in Fig. 1b.

- The detail in panel c is very small, so that the various symbols given are not clearly distinguishable. This figure may need enlarging.

We agree with the reviewer and added Type I and Type II to Fig. 1b.

We respectfully disagree with the reviewer's assessment of panel c. In our view, panel c effectively communicates the intended information.

p3:

A little more explanation of RESI may be good, but replacing "separating" with "labeling" may be sufficient.

We changed the sentence accordingly: "After mAb treatment and fixation, we performed stochastic separation by labeling with four orthogonal DNA sequences per target according to Fig. 1c."

p3: "Angstrom resolution"

While this is the headline from ref. 15, this isn't the case in all experiments, and it would be worth clarifying the situation in this experiment for the reader in the text at some point. ~0.6 nm s.d. precision is given in Fig. 1, which would give 1.4 nm FWHM if visualised with an appropriate spread for the uncertainty and would preclude at least resolution of targets separated by ~1.4 nm.

This is true, we reach approx. 1.4 nm FWHM in 3D measurements. We do not claim that we achieve Ångström resolution in our measurements. We believe that giving exact numbers for resolution based on the achieved precision in these specific measurements is not possible, since we are still limited by the label size. ALFA-tag and ALFA-nanobody are ~ 2nm in size and the GFP-tag and cognate nanobody are ~5 nm in size, limiting the achievable resolution. We now write: We achieved an effective resolution of ~2 to ~5 nm, limited by the size of the ALFA-tag-Nb or the mEGFP-tag-Nb complex, respectively."

p4: RESI precision – it would be good to mention that this precision is obtained when orthogonal labeling strands are on nearby proteins, so in a specific proportion of cases, but not every time.

We appreciate the feedback from the reviewer. We have now added a sentence to clarify this: "In order to achieve this, two adjacent molecules have to be labeled with orthogonal docking strands."

p4: "molecular resolution image"

I would prefer "an image resolving individual molecules", which would be more specific to this case, rather than a resolution claim that may sound more general, but I will leave this to the editors to decide on.

We appreciate the reviewer's perspective and have changed the sentence accordingly.

Fig. 2:

- Much of this is low contrast and hard to see.

- What is the scale bar for c, f?
- Depending on the scale of f, this is a nice demonstration of RESI.

We thank the reviewer for the helpful suggestion regarding Fig. 2. In response, we have enlarged panels c and f to improve clarity. We believe that this updated version offers improved visibility of the RESI clusters. Thank you for pointing out this issue, we now added the length of the scale bars to the figure captions of Fig.2c and f.

Sup. Fig. 1:

- Same contrast and visibility issue as Fig. 2

We appreciate the reviewer's suggestion to improve visibility in Sup. Fig. 1. Similar to the improvements in Fig. 2, we have increased the size of panels c and f in the revised figure. We believe this adjustment enhances the visibility of the RESI clusters and improves the overall readability of the figure.

p5: The axial distances may be more clearly determined and given, as suggested, rather than ~30 nm and "similar" distance.

Thank you for pointing this out. We now determine mean and std for the distances and have added this to the main text and figure caption.

Fig. 3:

- I appreciate the high-precision nature of the RESI localisations and the benefit of seeing a larger ROI, but the specks of colour are small again. I just highlight this for the editors.

We agree with the reviewer's assessment of the RESI localizations in Fig. 3 and have revised the figure accordingly by providing a more zoomed-in version of panels a and f to enhance clarity.

- I assume the grey circles and to do with the use of DBSCAN, but an explanation of these is needed.

We now added "DBSCAN cluster analysis" to panel a and f as well as "DBSCAN (clusters marked in gray)" to the figure caption to address this.

- Understanding the "normalization" in the Methods is essential here. Can the reader be taken through the hidden steps to adjust for labelling efficiency and antibodies per ALFA-Nb signal, to be convincing about this data? Seeing 0.5 OBZ per cluster is confusing to start with. Similarly, the red slope in panel b is not 1:2.

In the ratio analysis, we decided to treat 2 ALFA-Nbs within a cluster as one RTX or OBZ. Especially in the case of RTX, this nicely shows the linear correlation of RTX and CD20. Due to the labeling efficiency of 40% of the ALFA-Nb, this can result in 0.5 OBZ per cluster. We now clearly state this in the figure caption: "Two ALFA-Nbs correspond to one RTX/OBZ per cluster."

We agree that understanding the normalization is essential for the data. We already addressed this in the Methods "Ratio analysis" and now referred directly to this in the figure caption and main text.

We also added the sentence to the figure caption to describe the slope: “A linear fit yields 0.38, and corrected for the labeling efficiencies of the ALFA- and GFP-Nb, this suggests that approximately one RTX molecule binds per CD20 dimer (for details on the calculation, see Methods).”

- 3g, i: The ratios in g do not appear to match the ratios in i. Why is this?

The ratios of 3g are directly determined from the analyzed data and are not labeling efficiency corrected. The proportions of complexes in 3i are determined from simulations taking into account the labeling efficiency. We now state this clearly in the figure caption: “Simulations with CD20 monomers, dimers, trimers and tetramers, taking into account the labeling efficiency of 50 % of the GFP-Nb, for this representative cell result in 35 % monomers, 48 % dimers, 10 % trimers and 7 % tetramers after OBZ treatment.”

A new supplementary figure (or two) showing the simulations for Fig. 3, Ext. Data Fig. 2 and Fig. 4 would be very helpful.

We now added one supplementary figure displaying the data of the chain simulation for RTX treatment and the oligomer simulation for OBZ treatment (Fig. 3, Ext. Data Fig. 2) and one supplementary figure displaying the oligomer simulation for i-TCE and c-TCE treatment of Fig.4.

Suppl. Fig. 2: What data is this figure from? It seems very close to that of Fig. 3 (except for Suppl. Fig. 2c – is the y-axis mis-labelled?). It is some kind of repeat?

The data of the Suppl. Fig. is a replicate, we now indicate that in the figure caption. We changed the y-axis label in Suppl. Fig. 2c to the correct values.

p6: Molecular ratios – A note somewhere on how the presence of some pairs of proteins with the same RESI labeling strand affects this or does not affect it would be helpful.

The presence of some pairs with the same docking strand can lead to an underestimation of higher-order oligomers, if all of them are DNA-PAINT unresolvable. However, due to a high flexibility of the labeling probe, not all molecules within an oligomer are DNA-PAINT unresolvable so that this effect is mitigated. We assessed this in the RESI paper [<https://doi.org/10.1038/s41586-023-05925-9>] and now mention this in the main text. “Even though the detection efficiency of oligomers in RESI may be reduced due to two adjacent DNA-PAINT-unresolvable molecules labeled with the same DNA docking strand sequence, this cannot explain a trimer and tetramer proportion below 30 % [<https://doi.org/10.1038/s41586-023-05925-9>].”

p6: Distances – I assume these are from projections onto the XY plane, and Z information has been disregarded here, both in measurements and simulations. That seems reasonable, but it would be good to specify.

That is correct, the nearest-neighbor distances are quantified in 2D. We have now specified this in the figure caption.

p6: Comparison with CSR distribution: What intensities were used in the CSR simulations (Ext. Data Fig. 2), and how were they chosen?

The CSR distributions are simulated at the same molecular densities that were measured in the cell images, as specified in the methods “Simulation of monomers”.

p6: Chain segment length ~23 nm – Can this be given with an uncertainty again? A fit of the 2nd NND distribution may be another option?

We now performed Monte-Carlo simulations to assess the uncertainty of the chain segment lengths and show this in the same Extended data figures as before.

p6: C1q platforms

More tentative language seems appropriate, for presenting a new hypothesis when binding of C1q is not included in the study, and the specific shape of the actual chains is not extracted or described.

e.g.

- “the flexible nature of RTX-CD20 chains may explain”

- “For instance, U-shaped chains... could position ≥ 6 ... in close proximity”

We now changed the text to the following:

“Furthermore, the flexible nature of RTX-CD20 chains can explain how RTX assemblies organize to form C1q binding platforms.”

“For instance, U-shaped chains of 6 CD20 dimers could position 6 RTX-Fc domains in close proximity to allow for efficient C1q binding.”

- The Methods specify that the same ALFA-Nb may (usually?) be used in more than one overlapping C1q platform in Sup. Fig. 2b, to represent increased avidity. This needs stressing in Fig. 2, Sup. Fig. 2 and Ext. Data Fig. 2(h) as well.

We now added a sentence in the Fig.3 that clearly states that all binding possibilities are counted as C1q platform. “The number of hexameric RTX platforms was determined by counting each possible binding configuration of C1q, in order to represent the apparent gain in avidity, thus individual RTX molecules can be part of several distinct platforms.”

- Can the authors say anything about whether C1q requires the antibodies to be in a strictly hexagonal arrangement or not (comparing with ref. 20?), and compare with more detail of the ALFA-Nb localisation pattern?

Unfortunately, we cannot directly deduct hexagonal patterns from the data. The flexibility of both IgGs and ALFA-labels prevents that. C1q flexibility as well as IgG flexibility suggest that the hexagonal arrangement is not the most important feature but rather the proximity to increase avidity plays a role.

p7:

- CSR simulations – how was the parameter for this chosen in each case?

CSR simulations were performed as explained in the Methods “CD20 low order oligomerization simulations” by simulating only monomers that are CSR distributed.

- NND distances – what were they, quantitatively? These peaks are not very clear in the histograms, more in the simulation results.

They were below 20 nm. We now mention this in the main text: “This comparison revealed specific peaks at distances below 20 nm for 1st to 3rd NNDs, which cannot be explained by a pure CSR distribution.”

- Few compatible C1q binding platforms were detected for OBZ, not none.

We changed it to: “Accordingly, we did detect only a few compatible C1q binding platforms for OBZ (Extended Data Fig. 3h) as compared to RTX.”

Membrane bending:

- Can CD20 be tilted with respect to the membrane, rather than the membrane being bent?

Yes, that can happen, we changed the text accordingly to: “...suggesting local nanoscale bending of the cell membrane, although we cannot exclude that tilting of the whole complex contributes to this phenomenon.”

p8: TCEs and flexibility

Why should the i-TCE (anti-CD3 Fab between OBZ Fabs) be expected to be more flexible than the c-TCE (anti-CD3 on the end of OBZ Fab), as is given as reasoning?

The flexible linkers between anti-CD3-Fab and anti-CD20-Fab as well as the presence of anti-CD3-Fab between the two CD20-Fabs in inverted TCE (i-TCE) allow for a higher degree of freedom of CD20 binding when compared with the classical TCE (c-TCE).

Fig. 4, Ext. Data Fig. 4, Suppl. Table 2: How the data in the table results in the plots is not clear.

We now refer to the Supplementary Table 2 in the Figure captions of Fig. 4 and Extended Data Fig.4 (now 6). These correspond to the different sheets in the excel table, which we now explicitly write out in the caption.

p9: OFA distances

- Distances are given as exact (17 nm, 23 nm); estimates and uncertainties are needed.

We performed Monte Carlo simulations to estimate the uncertainties of the chain-model for different distances. The results are now plotted in Extended Data Fig. 3 for RTX and Extended Data Fig. 7 for OFA.

Ext. Data Fig. 5: Panels c and d are described the other way around in the caption.

We apologize for this oversight and changed the text accordingly.

- About the C1q platform hypothesis:

"...while also forming _possible_ C1q platfoms..." or similar.

We changed the text to "...while also forming platforms compatible with C1q binding."

p11:

- TCEs: Results in the presence of T-cells are mentioned, but the relationship to this experiment in different cells needs explaining.

*We changed the text: "Interestingly, when only investigating the CD20-mediated direct cytotoxicity, **independent of T cell mediated effects**, we found that **i-TCE** reduces the original direct cytotoxicity of **c-TCE** while it increases the formation of CD20 trimers and tetramers."*

- "The inverted format", last sentence, para 1. Is this the classical format – the inverted format of the inverted format? It reads as if it is different from i-TCE in the previous sentences.

*Sorry for the lack of clarity in the text. We changed the text to: "Interestingly, when only investigating the CD20-mediated direct cytotoxicity, **independent of T cell mediated effects**, we found that **i-TCE***

reduces the original direct cytotoxicity of c-TCE while it increases the formation of CD20 trimers and tetramers.”

-- If so, the “partial increase in CD20 oligomer formation” was assessed statistically but not found to be statistically significant, so care should be used in stating this.

This should be solved now based on the previous clarification.

- para. 2, sentence 1 on pathways – Personally, I would have found something like this helpful as extra background in the introduction.

We think it is appropriate to mention this only in the discussion to keep it as concise as possible.

- “Only with RESI...” – The only comparison is with DNA-PAINT, which should be made clear. However, it has also not been shown that these findings would not have been possible from the DNA-PAINT data (some kind of demonstration of this is suggested in initial comments, e.g. equivalent to Fig 1c,h from DNA-PAINT data).

We added Extended Data Fig. 2 to assess the advantage of RESI over DNA-PAINT.

- Other high-precision techniques (e.g. MINFLUX, ...FLUX) may also be discussed with advantages and drawbacks.

We now added 2 paragraphs to extensively discuss advantages and drawbacks of dSTORM, MINFLUX DNA-PAINT and RESI (see above).

- “revealing that Type I mAbs can form C1q platforms in chain-like arrangements...” – this has only been assessed without C1q in place and without explaining any restrictions on the geometry of the arrangement required for C1q - I think this should be more tentative, as above.

We changed it to: “...revealing that Type I mAbs can form platforms compatible with C1q binding within chain-like arrangements...”

- “the shorter chain length we measured” – I think “measured” is a bit strong after all of the simulations and picking the point with least disagreement; something like “deduced” might be ok.

We changed it to “deduced” accordingly.

p12: >10 cells per day... a powerful tool for screening

- This sounds like a low number for a powerful screening tool. Can the authors explain why this makes it a powerful screening tool, or describe it differently?

We do agree, however we believed that this is currently the most powerful screening tool at this resolution. Thus we changed the text accordingly: "The current throughput of RESI allows imaging >10 cells per day, making it a powerful tool for screening therapeutic mAb candidates at molecular resolution."

p17: CD20 chain-like simulations

- What was the "maximum number of chain segments" and how was it chosen?

It was chosen according to the experimentally determined cluster size. Each chain was simulated separately with a length according to the frequency of experimentally determined cluster sizes. For details see Methods "chain simulation".

Ext. Data Fig. 4 caption:

- "approaching values for" does not seem like the right phrase in either case when the i-TCE results are closer to OBZ and c-TCE than to RTX

We changed it to "Trending toward values for" where appropriate.

Ext. Data Fig. 5g:

- The histogram bins should be centered on integer values, so the reader can understand the data.

We changed the Extended Data Fig. 5 (now 7) accordingly.

Reviewer #1 (Remarks to the Author):

All of my comments have been adequately addressed. In my view, the manuscript is ready for publication, pending correction of a small number of typographical errors:

The newly added sentence „Importantly, Fc-Fc interaction blockade does not change the structural RTX-CD20186 arrangements observed with RESI, suggesting that this RTX-mediated CD20-clustering and the formation of C1q-187 binding platforms is independent of RTX Fc-Fc interactions (Extended Data Fig. 4)“ is missing a full stop at the end.

We thank the reviewer for noticing the mistake and we now added a full stop.

The caption of Extended Data Fig. 4c should read „SDS-PAGE of negative control for unspecific binding of BSA and IgG1 to Ni-NTA beads. ...“

We thank the reviewer for noticing the mistake and we changed the sentence accordingly.

There should be a blank before the caption of Extended Data Fig. 4f, at the moment it reads „... flexible-chain-like arrangement.f, Quantitative analysis ...“

We thank the reviewer for noticing the mistake and we now added a blank.

Reviewer #2 (Remarks to the Author):

I appreciate that the Authors have added the additional sentence regarding my comment. I have no further issues to mention and recommend publishing the study.

We thank the reviewer for the recommendation to publish our work.

Reviewer #3 (Remarks to the Author):

The authors have made many helpful improvements to the manuscript.

We thank the reviewer for the appreciation of our manuscript.

A couple of further minor changes would be beneficial before publication:

The new discussion related to STORM, DNA-PAINT and MINFLUX:

I agree with this discussion, however, it seems longer than is needed.

- It could perhaps start from "MINFLUX and related techniques achieve localization precisions of ..."

We agree and we changed the discussion accordingly.

- "~100² μm² fields-of-view" - this seems relatively unusual (although true) notation for field of view size (e.g. ~100 x 100 μm), and this sentence including the parenthesis at the end is confusing (an acquisition rate would be expected following throughput, and the parenthesis may be redundant).

We thank the reviewer for pointing that out and changed the notation accordingly. We also removed the parentheses to avoid redundancy.

- The sentence afterwards ("Thus, RESI features a dynamic range...") would be better reconnected with the sentence before the new discussion.

We agree with the reviewer and changed the discussion accordingly.

- I recommend considering the flow of the whole Discussion section in refining this.

We thank the reviewer for pointing this out. We hope to have improved the flow of our manuscript.

Uncertainties, Extended Data Fig. 3e and Extended Data Fig. 7c:

This new analysis is appropriate, but the definition used for "[statistically] significant difference" in the Methods and the definition of the asterisks in the plot should be included clearly.

We now added the definition of the vertical lines, error bars as well as the asterisks in the figure captions:

"Centered vertical lines represent the means, and the error bars represent the standard deviations. Statistical significance was assessed using an unpaired t-test."